# Combined genetic and chemical methods boost the precision of tracing illegal timber in Central Africa

Laura E. Boeschoten [1,2,13] ✉, Barbara Rocha Venancio Meyer-Sand [1,3,13] ✉, Arnoud Boom [4], Gaël U. Dipelet Bouka[5], Jannici C. U. Ciliane-Madikou[6], Nestor L. Engone Obiang[7], Mesly Guieshon-Engongoro[5], Arjen de Groot[7], Joël J. Loumeto[5], Dieu-merci M. F. Mbika[5], Cynel G. Moundounga[8], Rita M. D. Ndangani[5], Dyana Ndiade-Bourobou[9], Ute Sass-Klaassen[1,10], Marinus J. M. Smulders [11], Steve N. Tassiamba[12], Martin T. Tchamba[12], Bijoux B. L. Toumba-Paka[5], Mart Vlam[1,10], Herman T. Zanguim[12], Pascaline T. Zemtsa[12] & Pieter A. Zuidema [1]

Enforcement of national and international laws banning illegal tropical timber trade hinges on independent origin verification, such as with genetic or chemical wood properties. This is of particular concern in Central Africa, where illegal trade prevails. However, tracing methods have not yet consistently achieved high accuracy (>90%) at small spatial scales (<100 km). Where high precision is required but individual methods fall short, combining methods may improve results, because drivers of wood properties differ. Here, we assessed the individual and combined identification potential of three methods (genetics with 238 plastid Single Nucleotide Polymorphisms, 3 stable isotopes, and 41 elemental concentrations). The combined approach achieved unprecedented accuracy in Central Africa, identifying 94% of samples within 100 km of their origin, outperforming individual methods (50–80%), and verifying real origin for 88%. These findings show that method complementarity boosts tracing accuracy and spatial precision, crucial for high-value timbers or high-risk regions.

Illegal logging damages forest ecosystems, undermines the sustainable management of forests, reduces tax revenues, and contributes to global crime. Estimates of the associated illegal timber trade range from 8% to 29% of the traded volume globally[1], making it the third largest transnational crime worldwide[2,3]. It is a persistent environmental problem, especially prevalent in tropical countries[1]. For example, up to 50–90% of exported timber is estimated to be from illegal sources in Central Africa[1]. To combat the trade in illegal timber, national and international legislation demands species and origin declarations for traded products. Examples of such legislation include the US Lacey Act, the UK Timber

Regulation, and the Australian Illegal Logging Prohibition Act. The European Union goes beyond legality: the new Regulation on Deforestation Free Products (EUDR, planned to take effect on Dec 30, 2025) mandates importers and traders to ensure that timber is both legally sourced and does not contribute to deforestation. To comply with the EUDR, importers are required to provide exact geolocations of timber and thereby prove that it is sourced from areas that were not deforested after the cut-off date[4]. A major challenge, however, in the enforcement of the various legislations is the possibility to independently verify claimed timber species and origin[5,6].

[1]Forest Ecology and Forest Management Group, Wageningen University and Research, Wageningen, The Netherlands. [2]Department of Ecology, Evolution and Environmental Biology, Columbia University, New York, NY, USA. [3]Forest and Nature Conservation Policy Group, Wageningen University and Research, Wageningen, The Netherlands. [4]Department of Geography, University of Leicester, Leicester, UK. [5]Laboratoire de Biodiversité, Gestion des écosystèmes et de l'Environnement, Faculté desSciences et Techniques, Université Marien Ngouabi, Brazzaville, Congo. [6]Herbier National du Gabon, Institut de Pharmacopée et de Médecine Traditionnelle (IPHAMETRA), Centre National de Recherche Scientifique et Technique (CENAREST), Libreville, Gabon. [7]Wageningen Environmental Research, Wageningen University and Research, Wageningen, The Netherlands. [8]Institute for Research in Tropical Ecology (IRET/CENAREST), Libreville, Gabon. [9]Institute for Agronomic and Forestry Researches (IRAF/CENAREST), Libreville, Gabon. [10]Forest and Nature Management, Van Hall Larenstein University of Applied Sciences, Velp, The Netherlands. [11]Plant Breeding, Wageningen University and Research, Wageningen, The Netherlands. [12]Laboratory of Environmental Geomatics, Department of Forestry, Faculty of Agronomy and Agricultural Sciences, University of Dschang, Dschang, Cameroon. [13]These authors contributed equally: Laura E. Boeschoten, Barbara Rocha Venancio Meyer-Sand. ✉e-mail: leb2239@columbia.edu; barbara.rvmeyersand@wur.nl

Multiple scientific identification methods are already successfully applied in forensic cases to catch species false declarations. These species identification methods include wood anatomy[7], genetics[8], Near-Infrared Spectroscopy[9], DART-TOFMS[10,11] and stable isotopes[12]. While challenges remain in identifying species within complex genera with high similarities[13],[14], these scientific methods offer the practical toolbox to achieve species identification with high accuracy[6]. In stark contrast, the verification of origin claims still heavily relies on external documents, tags, or scans, leaving room for fraudulent practices. Even though various forensic methods are being developed and some have shown good tracing results, they are still not widely implemented. None of the three currently used methods (genetic analyses, stable isotope ratios and multi-element analysis) has consistently demonstrated high accuracy and spatial precision to trace the origin of tropical timbers. In some cases, the results have been promising[15–19], but in others, there was insufficient environmental and/or genetic variation present in the study area to provide accurate tracing at small scales[5,6,12,19,20].

Evaluating tracing performance also depends on the tracing question at hand, as tracing methods need to work at a relevant accuracy and spatial scale for law enforcement[21,22]. Here, we define these as an accuracy above 90% to distinguish between origins at a spatial scale of below 100 km,

allowing origin verification at a sub-country scale. Yet, the required accuracy is highly context dependent. In forensic cases with ample availability of other types of evidence, a lower accuracy of timber tracing may be sufficient to build a legal case. Furthermore, not all cases require the distinction between sites as close as 100 km apart. However, when tracing at the scale of 100 km could be conducted with high reliability, this would allow for the identification of many prevailing timber laundering routes[3,23].

When high-resolution tracing is required, combining methods may improve tracing accuracy, as different methods can provide complementary information on wood origin. Evidence for such methodological complementarity in origin determination has been provided for various agricultural commodities[24], archaeological wood samples[25,26] and temperate timber[27]. Yet for tropical timbers, the potential of combining tracing methods has not been quantified. Central Africa specifically presents a highly relevant case for this type of study for multiple reasons: the levels of illegal timber trade are persistently high[1,28], geographical barriers that would induce strong genetic isolation are scarce and the spatial variation in climate, topography and bedrock or soil type is rather limited (Fig. 1)[20]. This geographic homogeneity limits the potential for individual tracing methods at small spatial scales, but offers prospects for complementarity of methods to

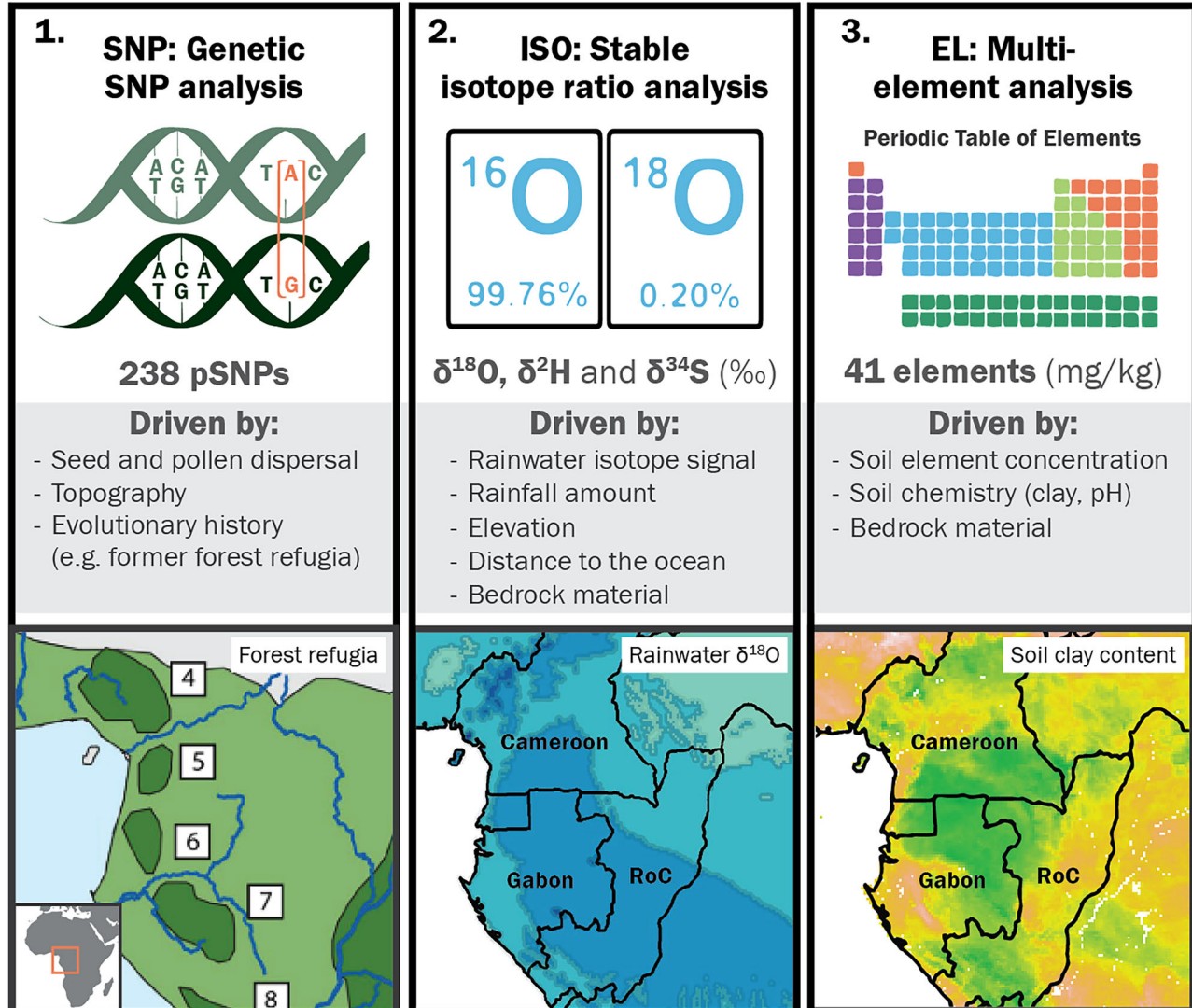

**Fig. 1 | Overview of the three methods for timber tracing in this study and their drivers of geographic variation.** Possible drivers of geographic variation shown as examples are: 1. For SNPs: former forest refugia during cold Pleistocene glaciation cycles with refugia in dark and forest cover in light green, adapted from Allen et al.[81], 2. For stable isotopes: the oxygen isotope signal in rainwater, with higher rainwater δ18O in lighter blue[82] and 3. For multi-element analysis: soil clay content ranging from high clay % in green, mid-levels of clay % in yellow and low clay % in pink[83].

reach adequate levels of tracing accuracy and precision. For these reasons, we tested genetic, isotopic, and elemental tracing methods to evaluate their combined potential in Central Africa.

Genetic tracing methods rely on spatial genetic structure, a genetic mosaic composed of the variation of genetic profiles across the landscape. This genetic mosaic is shaped by reproductive, demographic and historical biogeographic factors, such as seed and pollen dispersal, topographic barriers like rivers and mountain ranges, and evolutionary history, including glaciation cycles during the Pleistocene that led to forest refugia[29] (Fig. 1.1). For example, seed and pollen dispersal strategies  can lead to more similar genetic profiles in trees in close proximity than those of trees further apart (isolation-by-distance pattern). Genetic methods are widely studied and employed for tracing[6,30], with potential to differentiate the origin often at the region (e.g., West- versus Central Africa) and country levels[31], and have yielded high tracing accuracy at short distances in some cases[19,32,33]. Stable isotope ratio analysis (mostly bio-elements $\delta^{18}O$, $\delta^2H$, $\delta^{13}C$, $\delta^{15}N$ and $\delta^{34}S$) uses spatial variation in wood chemistry generated by climatic conditions, geology and atmospheric deposition[34–42] (Fig. 1.2). This technique is frequently applied in timber tracing, and has shown best accuracy at large spatial scales[19,35,37,43] and in areas with large isotopic variation such as in mountain ranges[34]. However, recent developments have shown good results when applied in spatial prediction maps[44] as well as in combination with elemental analysis[27]. In Central Africa, results have been variable, with one study showing isotopic differences between two sites[15], but another study not showing high potential for site identification at a sub-country level[37] using data from 17 sites. The third method, multi-element analysis[18,20,27], is rather new and implies measuring a large number of elements in wood (such as Mg, Ca, La). Geographical variation in the resulting wood elemental composition is mainly caused by variation in physical and chemical soil properties (Fig. 1.3)[18]. This method has shown high tracing accuracy at small spatial scales (50–100 km), but performs less well to differentiate distant origins[20]. For temperate tree species in Eastern Europe, a smaller set of elements showed good origin verification results when combined with stable isotopes[27].

Here we tested the potential of combining three forensic methods - genetics, stable isotope ratios and multi-element analysis - for the identification of tropical timber origin in Central Africa. We applied these methods to Azobé (*Lophira alata* Banks *ex* C. F. Gaertn.), a flagship species listed as vulnerable by the IUCN. Azobé wood is very dense and durable, primarily used for road and waterworks due to its exceptional strength and resistance to decay[45] and is heavily traded from the Congo Basin. Trees were sampled across 13 locations in the main Azobé exporting countries: Cameroon, Gabon, and the Republic of the Congo, covering a range of distances. The two closest sites were 15 km apart, whereas the two furthest sites were over 1000 km away. The genetic database was developed by the detection of plastid (chloroplast) genome-wide single-nucleotide polymorphisms (pSNPs)[46], building on the idea proposed by Li et al. (2015) of using the chloroplast genome as a super-barcode[47]. pSNPs for each tree were combined in individual genetic profiles. The isotopic database included measurements of $\delta^{18}O$, $\delta^2H$ and $\delta^{34}S$[37]. Lastly, the multi-element database comprised of 41 elemental concentrations, including trace elements known to be used by plants as well as 16 rare earth elements (see Methods for the full list). To evaluate the complementarity of the methods, we assessed the tracing potential per method and for all combinations of methods.

In this study, our primary focus is on origin identification, which aims to determine the specific source of the timber ('Where did this timber come from?'). This is related to, but also distinct from, origin verification[48], where the goal is to confirm whether timber originated from a specific location, e.g., the origin claimed in trade documents ('Did this timber come from location X?'). Testing the potential for origin identification is a crucial step in the development of forensic methods as it gives insight into how variation is distributed across different origins. Therefore, it is commonly used in the tracing literature to test the method performance[15,19,49,50]. However, in forensic cases, the more common question is that of verification. We therefore

also perform a test whether combining methods increases the verification success.

## Results
### Genetic analysis shows three distinct clusters and spatial structure
We first compared the geographical variation and performance of each tracing method. A total of 238 pSNPs for 234 individual trees composed the genetic dataset after quality checks and filtering, providing the basis for genetic tracing, as described by R. V. Meyer-Sand et al.[51]. Some known genetic splits include those across the Lower Guinean region, from 0°8' to 3°8'N latitude, and the Cameroonian Volcanic Line[17]. The RDA analyses (Supplementary Fig. S2) indicated three main genetic clusters based on the proportion of shared alleles among individual genetic profiles. The first main cluster consisted of 10 sites, spanning from West Cameroon, to North-West Congo and Central-East Gabon. Within this cluster, two sites in West Cameroon (CAM1 and CAM3) were distinct from the others (Fig. 2A). The second cluster in West Gabon was the most genetically distinct and may be composed of a cryptic *L. alata* species (L. alata1 - West Gabon[52]). The third cluster was located in Northern Congo (Supplementary Fig. S2) and represents a genetic group that previously remained undetected using nuclear micro-satellite markers[52].

Genetic Random Forest classification models yielded correct identification to the site of origin for 46.2% ± 3.3% of the samples (Figs. 2A, 3), but values varied largely across sites. Success rate varied from 0% at some sites to 100% at others (Fig. 2A). The genetic groups exhibited a clear spatial structure: incorrect identifications often occurred between groups of sites that were geographically close (such as CON2 and CON4). This was also clear from the distances to predicted origin: 62.2% of the trees was assigned within 100 km of the sample site, 85.6% within 300 km, and no trees were assigned to sites more than 500 km away (Fig. 3). In short, genetic tracing combined a relatively low accuracy of assigning the correct site, with a high precision to assign within relatively short distances.

### High local variation in stable isotope ratios suggests limited potential
Across sites and trees, stable isotope ratios ranged from 24.7–30.4 ‰ for $\delta^{18}O$, −9.5–36.55 ‰ for $\delta^2H$ and 4.6–10.9 ‰ for $\delta^{34}S$. Local variation in isotopic ratios of trees was high, which resulted in a standard deviation within the sites that was similar to the variation in isotope ratios across all sites (Supplementary Table S1). However, even with this high isotopic overlap among sites, site identity was significant in the db-RDA analysis ($p < 0.001$, Supplementary Fig. S3). Especially $\delta^2H$ was a strong predictor for the first axis, which explained 61.5% of the total variation.

Despite the significant differences among sites, identification accuracy was 40.7% ± 7.0% (Figs. 2B, 3). There was no clear spatial structure in the assignments, which can be seen in Fig. 2B by comparing the colours of sites that are located close by and far away. Trees were assigned to many different sites (up to eight out of 13), and only 49.8% of the trees were assigned within 100 km. Misassignments occurred as far as 1000 km from the original origin (Fig. 3), indicating low precision.

### Multi-element analysis shows high identification accuracy
Concentrations of the 41 measured trace elements varied between 0.001 g/kg (Yb) and 4.0 g/kg (K) in the wood samples. Differences between sites were clear, causing the site to be a significant predictor in the db-RDA ($p < 0.001$, Supplementary Fig. S4). Site differences were mostly resulting from distinct values of a few trace elements, similar to a previous study on a larger number of sites using this method[20]. The Random Forest classification model based on the multi-elemental concentration performed better than that of the other methods: 73.4% ± 5.7% of trees were assigned to the correct site of origin (Figs. 2C, 3) and 81.3% of the trees was assigned within 100 km of its origin. Important trace elements for the classification based on variable importance were W, Ba, Mo, K and Cr. An identification accuracy of 100%

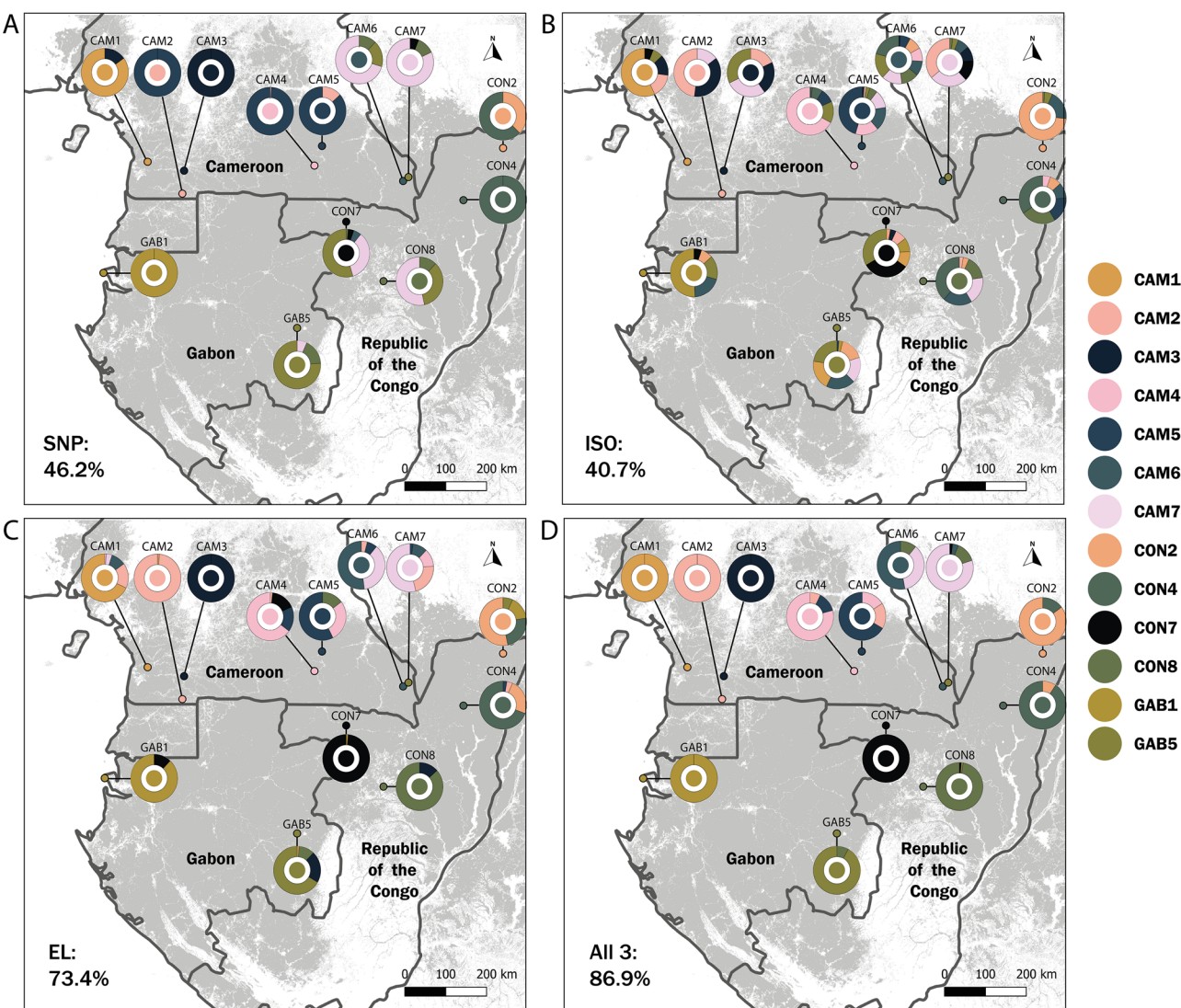

**Fig. 2 | Confusion charts for the identification of wood samples to their respective origin.** Identification was based on **A** pSNPs (SNP), **B** three stable isotope ratios (ISO), **C** multi-element analysis of 44 trace elements (EL) and **D** all three combined. Mean identification accuracy across all sites is indicated in the bottom left. Each site has a unique colour, shown in the inner circle and in the legend. Colours in the outer circle of each symbol indicate to which (other) site(s) the trees of that site were assigned. Primary tropical forest extent from Global Forest Watch is indicated in grey[84].

was reached at some sites (CAM2 and CAM3). However, some other trees could be assigned to sites as far as 1000 km away, indicating a low precision (Fig. 3). This result contrasts with the strong spatial structure in the pSNPs data, for which distances were much shorter to the correct origin of the misassigned individuals.

**Combining methods boosts identification accuracy and precision**

All pairwise combinations of the three methods increased identification accuracy compared to the individual methods (Figs. 2D, 3), indicating that the methods indeed complemented each other. Especially combinations including multi-element analysis resulted in a high accuracy (79.8% and 77.3% for combinations with genetic and isotopic data, respectively). These combinations also yielded the most consistent results, as indicated by a low standard deviation across the Random Forest models (Fig. 3). The combination of all three methods improved site identification accuracy even further, to a correct assignment of 86.9% ± 3.5%. It also yielded high spatial precision: 91.0% of all trees were assigned within 50 km of the sample site and 94.5% within 100 km. Furthermore, using the combination of methods, no trees were assigned more than 500 km away from their correct origin (Fig. 3).

The complementarity of the methods is illustrated in the charts of the site-specific assignments (Fig. 2). The set of sites exhibiting the highest accuracy for individual methods differed across methods. For example, trees from site CAM5 were very well identified genetically but not chemically, while sites CAM2 and CON7 could only be separated using elemental analyses, and site CON2 was best identified with stable isotopes. This resulted in clear methodological complementarity: regions where elemental analyses performed less well coincided with those where other analyses did well. For example, CAM1 and CON8 showed 100% correct site assignments based on the combination of methods, whereas none of the individual methods could distinguish these sites well.

**A potential tracing scenario: origin verification**

In practical tracing applications, verifying a claimed origin is more common than identifying an unknown origin from a set of possible sources. While these two scenarios are closely related, they present distinct challenges[27,48]. To evaluate the performance of our methods for origin verification, we designed a scenario in which a subset of 41 test samples was randomly assigned an origin claim from our set of 13 sites. Test samples were removed from the dataset, and verification models were built to assess performance for two verification tests, both common in forensic cases: (A) correct

**Fig. 3 | Distance to predicted origin (km) for different tracing methods and their combinations.**
Depicted as % of the total number of test trees per distance bin per method. The lowest bar represents correct site origin assignment, which is the identification accuracy, also represented by a dot and error bar that indicates the variation (st. dev.) in identification accuracy, estimated using different test and training datasets of the Random Forest models. Methods are faceted by analyses based on a single method (1), a combination of two methods (2) or all three methods (3). SNP = pSNPs, ISO = stable isotope ratios, EL = multi-element analysis, ALL = all three methods, RDM = the occurrence of pairwise distances in the reference dataset, representing the expected result of fully random assignments. The dotted line indicates 90% accuracy. A version of this figure including standard errors for all distance bins is provided as Supplementary Fig. S5.

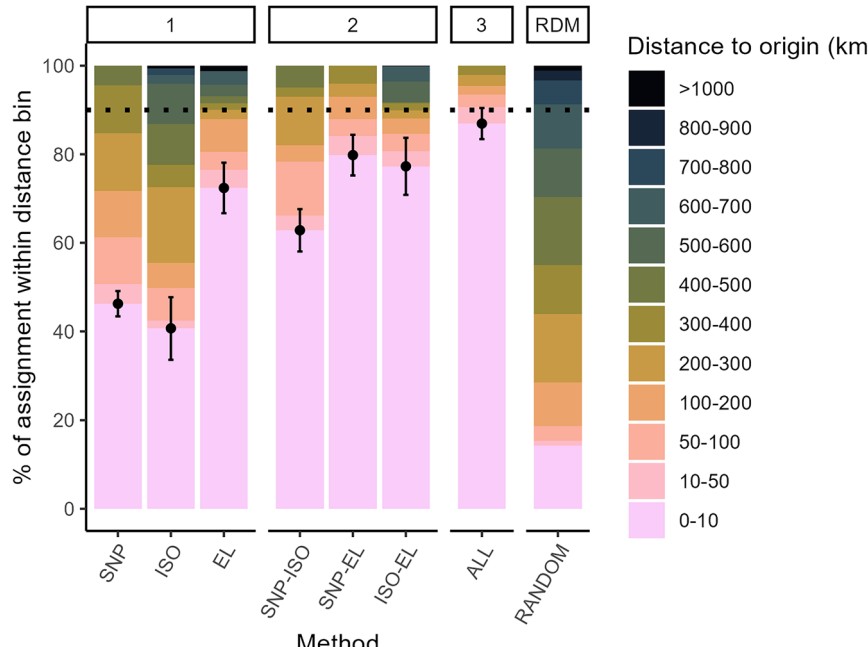

verification of true origin claims, and (B) correct rejection of false origin claims (see scheme in Fig. 4). To simulate a scenario where the claimed origin is correct (A), we tested whether verification models correctly assigned the 41 trees to their true origin. This reflects a situation where training samples can be collected from a claimed origin. Contrastingly, to simulate a scenario where the claimed origin is incorrect (B), we tested whether verification models correctly rejected a false origin claim for the same 41 trees. For these models, we excluded the other trees of the true origin from the reference database to represent a situation where the true origin is not the claimed origin, and moreover, the true origin is likely unknown and thus not present in the database.

Our results show that, consistent with those for the identification analyses, combining methods improves the accuracy of verification. The highest rate of correctly verified claims (scenario A) was achieved when all three methods were combined, with 87.8% of test trees accurately assigned to their true origin. These assignments represent the true positives, and these are crucial to ensure no timber traders are wrongly accused of claiming a false origin. For scenario B, all methods and combinations of methods were effective at rejecting false claims, with pSNPs achieving the highest correct rejection rate at 95.4%. These assignments represent the true negatives, which are important to find false claims.

For these verification models, it is particularly important to compare model outcomes to those expected from random assignment, as illustrated by the rightmost bars (RDM) in Fig. 4. In scenario A, where each tree is assigned to one of 13 possible sites in the reference dataset, the probability of a correct assignment by chance is only 7.7%, and all models clearly outperform this random baseline. In scenario B, where reference samples from the true origin are removed and trees are assigned to one of the remaining 12 sites, the probability of correctly rejecting a false origin claim by chance is 91.7%, since 11 of the remaining 12 sites represent the non-claimed origin, thus assigning a sample to those sites results in 'correctly rejecting' the claim. In this scenario, our models perform similarly to random assignments, reflecting the high baseline probability of correct rejection.

## Discussion
Origin fraud in the timber trade remains widespread and to date, no tracing method has been able to consistently and reliably distinguish between origins at distances relevant for forensic cases across diverse contexts. Especially in regions with limited geographical barriers and environmental variation, such as Central Africa, reaching high tracing accuracy is difficult.

Our results reveal how different tracing methods that use wood properties, which are driven by fundamentally different processes (Fig. 1), complement each other. Combining methods, we reached an unprecedented tracing accuracy in this region with 94.5% of our wood samples assigned within 100 km of their origin and 91% within 50 km. Furthermore, when the methods were applied for a pairwise verification scenario, we again found that the combination of methods performed best to confirm true claims (87.8% correct) while also effectively rejecting false claims (90.2%).

### Why combining methods improves tracing accuracy
The three tracing methods yielded varying results in accuracy and spatial precision. These differences are likely caused by differences in the environmental drivers that generate spatial variation in genetic, isotopic and elemental wood characteristics (Figs. 1, 2, 4). The drivers differ in their spatial structure and scale, which generates the complementarity that was demonstrated by the considerably higher identification accuracy and precision when methods were combined. Specifically, we hypothesize that multi-element profiles added fine-scale granularity to the spatial structure, improving site assignments (<50 km) when combined with other methods. This likely originates from the small scale at which soil properties can vary (Fig. 1), which is reflected in the short distances at which variation in wood elemental concentrations occurs (Fig. 2C)[18]. This was also reflected in the verification models: multi-element analysis resulted in the highest percentage of correctly confirmed origins (Fig. 4). On the other hand, the genetic profiles improved the spatial resolution, because they reduced the distances of misassignments that occurred in the elemental analysis (Fig. 3). As there are no large barriers in our study area that would amplify distinct genetic patterns (Fig. 1), we observe a more gradual change in genetic variation between most of our study sites. Thus, the spatial variation in drivers created mosaics of chemical and genetic spatial variation, which differed at both coarse and fine spatial scales. When methods are combined, these mosaics may then turn out to be complementary in terms of overall patterns and granularity. Again, this was confirmed in the pairwise verification tests, where we found the best results when combining tracing methods (Fig. 4).

It is always complex to compare study results across geographical regions, study design, study species and different statistical approaches. Therefore, to put our findings in context, we first compare them to the results of other single-method studies. Compared to what has been reported for timber identification using genetic tracing, the reported accuracy yielded by combining methods is high. For instance, genetic studies on tropical

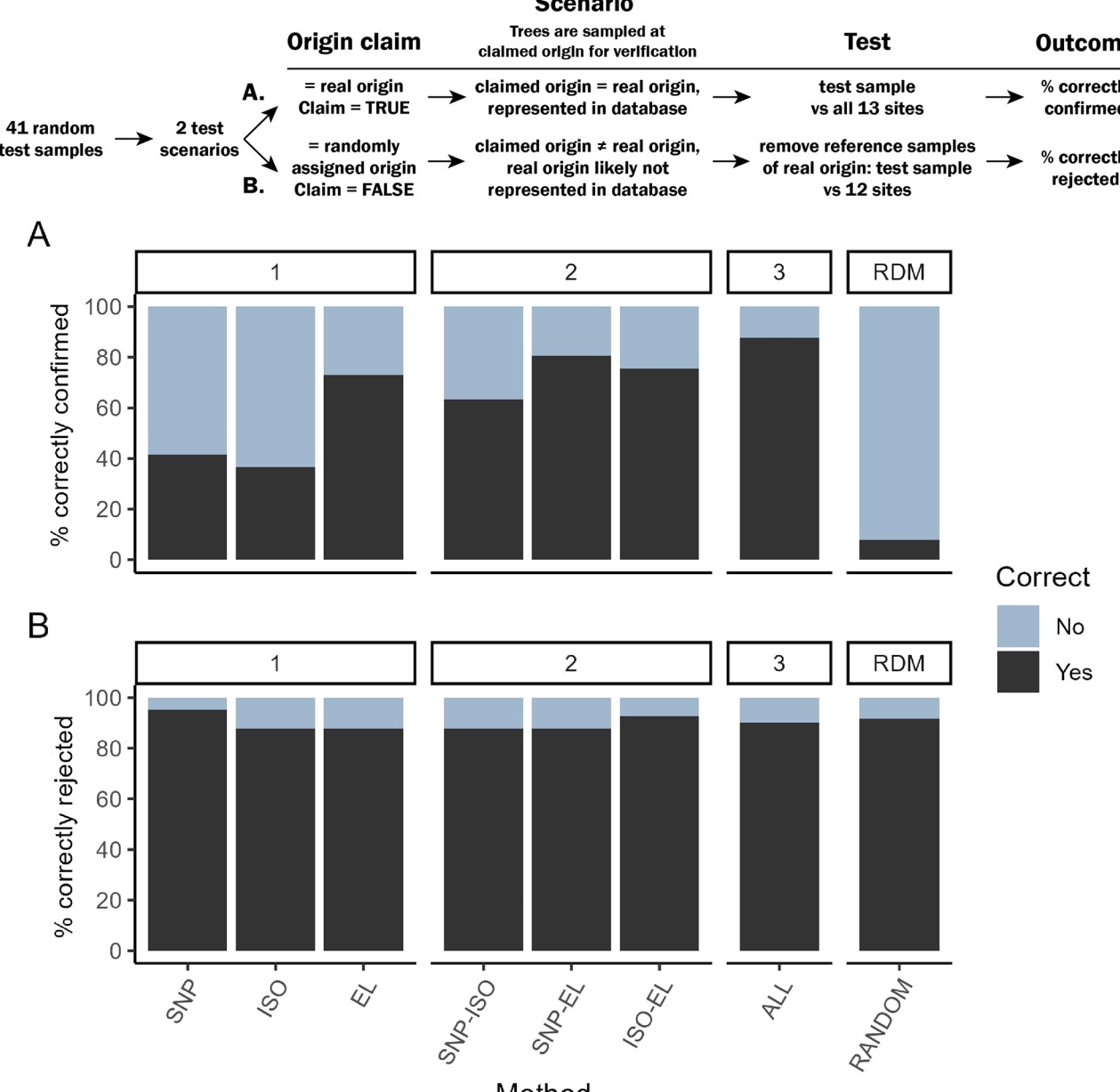

**Fig. 4 | Verification performance of single and combined tracing methods.** 41 test trees were randomly assigned an origin claim from one of our 13 sites. We then simulated two scenarios to calculate: **A** the percentage of correctly confirmed claims per tracing method or combination of methods, using the true origin (i.e., correct) as origin claim, and **B** the percentage of correctly rejected claims, where the randomly sampled claimed origin (i.e., incorrect) was used. For this second scenario, we removed the reference samples from the real origin of the test tree from the training dataset to mimic a tracing scenario. Methods are faceted by analyses based on a single method (1), a combination of two methods (2), or all three methods (3). *SNP* pSNPs, *ISO* stable isotope ratios, *EL* multi-element analysis, *ALL* all three methods, *RANDOM (RDM)* the percentage of correctly confirmed (A) or rejected (B) samples if assignment was completely random. For A: (1/13)*100% = 7.7%; for B (11/12) *100% = 91.7%.

timber yielded an identification success of 86% and 92% for 5 sites at 14–50 km[2] in Cameroon[17,19], but a considerably lower accuracy at distances between 10–300 km in Malaysia (60.6%)[50] and 268–501 km in Bolivia (66.3%)[53]. Isotopic studies on tropical timbers were unable to differentiate between sites at 14–216 km distance in Cameroon[19], within Cameroon, Gabon and the Republic of the Congo[37] or up to 400 km in Bolivia[12]. One study did find isotopic differences between two concessions in Gabon, but no tests were performed to quantify tracing accuracy[54]. At larger scales, distances or along elevation gradients, isotopic tracing studies did yield higher accuracy for temperate tree species[34–36]. For multi-element analysis, the only existing other timber tracing study was conducted for Tali timber in

an overlapping set of sites, yielding similar identification accuracy as for Azobé here (68.8% versus 72.6%)[20].

Our study constitutes a first test of the potential virtues of combining genetic and chemical tracing methods. The effect of combining methods on tracing accuracy and precision needs to be tested on other species and in other regions. A recent study did so for Eastern European timbers using stable isotope ratios and elemental composition, yielding high determination accuracy of harvest locations at 180–230 km of the true origin, but the authors used a different statistical approach, so a direct comparison of accuracy and precision is hard to make[27]. Our results suggest that the distance range in that study may be further reduced when adding genetic methods.

We anticipate that the added value of multiple tracing methods will be highest where the environmental gradients driving variation in wood genetic and chemical properties are small, but tracing at a high resolution is required. In these cases, individual tracing methods likely yield low accuracy and precision, and combining methods will then be valuable to improve tracing results. In contrast, individual methods may already perform good enough to answer the tracing question at hand in regions where physical barriers or environmental gradients are present, such as in mountain ranges[3] [4], making the investment in multiple forensic methods unnecessary. This holds not only for timber but could also apply to agricultural commodities and illegal wildlife[24,55–57]. Clearly, these hypotheses on the possible roles of geographic barriers and environmental gradients require further testing.

### Next steps to improve the performance of combined methods

To further improve small-scale timber tracing for Azobé in Central Africa, we suggest the following steps. For genetic analysis, the employment of a few nuclear DNA markers with high individual discriminating power in addition to the plastid DNA used in this study may add more genetic substructure to differentiate at the site level[31,58,59]. Additionally, tests are needed to quantify DNA quality obtained from dried wood samples, and compare these to the current dataset, which was developed from cambium samples. DNA extraction from wood remains challenging due to its low quantity and highly fragmented nature, but is possible even from processed wood[60,61]. Targeting chloroplast (cpDNA) or plastid DNA has been particularly powerful, as it is smaller and more abundant compared to the nuclear genome, making it easier to sequence than nuclear DNA[60].

To improve isotopic origin identification, the addition of carbon and strontium isotopes could further improve tracing results, depending on the region[27,62]. Additionally, our isotopic sample size was smaller than the other two methods. We had hypothesized, based on literature[27,34], that local variation would be limited and we could thus identify a site-specific isotope profile with fewer trees compared to elemental and genetic analysis. However, we did find quite a variation between trees at the same site. Therefore, measuring isotopic composition in a larger number of trees per site could have improved our isotopic profile. However, this would likely not have improved the performance of isotopic tracing in our study area because of the limited spatial isotopic variation[37].

Across all methods, it will be important to fill in the sampling gaps within the distributional area of Azobé. Even though the current sampling was extensive and covered the most important sourcing areas, various regions where Azobé is logged were missing from our reference database (central Gabon, Democratic Republic of the Congo). Initiatives for global wood sampling for timber tracing and centralized wood storage, such as those organized by World Forest ID[63], may help to fill these sampling gaps for Azobé. We think this will further improve the origin verification results presented here as well (Fig. 4), because the representation of more sites in the database will further improve the ability of our models to distinguish what makes a certain site unique. Finally, a blind sample test is a logical next step after this verification analysis to put the combination of methods to the test.

Beyond the case of Azobé, combined tracing methods may also be developed further by moving from identifying the site of origin, as we did here, to identifying likely regions of origin (origin determination)[36]. Such spatial predictions are ideally based on a larger number of sites than included here. Spatial prediction maps can be developed for SNP occurrence[64,65], isotope composition (i.e, isoscapes)[37,38] and elemental composition, once the underlying drivers are understood and their effects are quantified. By combining these predictive maps, the potential area of origin can then be defined for any blind sample with a specific accuracy[27]. An advantage of this map-based approach over the site-based approach employed here is that it does not require measurements for each method on all trees. Predictive maps can be developed separately per method and then combined. This would make methods more flexible and remove the need for overlapping reference datasets of all potential origins. A consequence of this approach is that it will not assign samples to their site of origin, but rather the potential region of origin and the probability that a sample came from a specific location[27,36]. Ideally, these maps would include prior information on driving variables in the area as well, such as rainfall or soil chemistry, to optimize predictions in areas without reference samples. Apart from a denser coverage of reference samples, this approach also requires further development of statistical methods, both for generating and combining the maps[36].

### When and where to combine tracing methods

Legislation to halt illegal, unsustainable trade will become more explicit about origin claims, such as the Lacey Act in the U.S.A., the UKTR in the United Kingdom[66] and the EUDR in the European Union from December 30, 2025[4]. As the demand for reliable origin verification of timber and other commodities like palm oil, soy, rubber, beef and cocoa continues to grow, so does the call for robust and reliable tracing methods. Deciding which (combination of) forensic tracing methods to use will depend on the spatial scale that is required to answer the legality/origin question at hand. Combining tracing methods comes at additional costs, time and effort, which should be considered when making this decision. The greatest advantage of the combined approach lies in its ability to overcome the limitations of individual methods when these yield insufficient accuracy and precision. Depending on the tracing question at hand, an assessment thus needs to be made, which method or combination of methods is the most feasible to identify the origin with high accuracy in the region in question. Higher accuracy through combining methods may be most warranted in high-risk areas such as the Congo Basin, where the share of illegal timber trade remains high[1], and individual methods do not achieve the required accuracy and precision. Furthermore, the additional effort of combining methods can be justified to trace the origin of highly valued wood, or wood obtained from threatened species, such as rosewoods, mahogany, ipê or teak, or to reconstruct illegal trading routes[67–69]. In such cases, investments in multi-method approaches may also pay off because publicity about effective tracing and court cases may encourage timber importers to carefully verify their supply chains and may prevent (more) operators from getting involved in illegal timber trade.

## Methods
### Sample collection

The study was conducted on the commercial Central African timber species Azobé (*Lophira alata* Banks *ex* C.F. Gaertn, Ochnaceae). Azobé occurs from Guinea to the Democratic Republic of Congo[70], but almost all timber export originates from Cameroon, Gabon, the Republic of the Congo and the Democratic Republic of Congo. Sampling was conducted in 13 sites, which were located in logging concessions in natural forest in Cameroon, Gabon and the Republic of the Congo between September 2019 and April 2022 (Fig. 2). All sampling was conducted in collaboration with the operating forestry companies. The sampling sites were at least 10 km apart. We named the sites using a shortened version of the country (CAM for Cameroon, CON for the Republic of the Congo, GAB for Gabon) combined with a number to indicate the site. We used the centroid of the latitude and longitude coordinates of all sampled trees within a site to represent its location.

At each sampling site, we sampled heartwood as well as bark of 20 trees. Target trees within one site were located between 100 m and 5 km apart: we did not aim to cover the whole concession; we rather aimed to characterize a population. Sampled trees were either standing or recently felled and were at least 30 cm in diameter at breast height (DBH). The size variation of the sampled trees was comparable across all sites. The heartwood sample was collected from each tree as an increment core (Haglöf Increment borer 350 mm × 5.15 mm), with a FAMAG plug cutter of 15 mm diameter, as a wood chunk or as a wood powder sample obtained with an electrical drill. All samples were taken at least 14 cm into the tree. The heartwood samples were stored in plastic straws or paper envelopes and properly ventilated to prevent mold. Additionally, three cambium samples per tree were taken with punches of 2.5 cm diameter. The cambium samples were stored in plastic bags with silica. The silica was refreshed up to 1 week after sampling

to ensure the samples were fully dried. Additionally, GPS-coordinates and DBH were recorded.

## Genetic analysis

The genetic lab methodology followed R. V. Meyer-Sand et al.[46]. In summary, DNA was isolated from cambium and leaf tissue samples from between 15 and 20 trees per site. Genomic material was isolated with an optimized cetyltrimethyl ammonium bromide (CTAB) protocol[71] with additional cleaning steps (Supplementary Methods S1). DNA purity of all extracts was checked with Nanodrop (Thermo Fisher Scientific, Schwerte, Germany), DNA concentrations were measured with the Qubit™ kit (Thermo Fisher Scientific, Schwerte, Germany) following the manufacturer's instructions, and a 1.5% agarose gel was used to check the fragments length range. The DNA isolates were used to prepare three libraries, 300 bp or more insert size, with the 'RIPTIDE High Throughput Rapid Library Prep Kit' (Twist Bioscience, South San Francisco, USA). The libraries were sequenced with Illumina Novaseq6000 PE150 (Novogene, Cambridge, United Kingdom). The Illumina sequences of the trees were mapped to the annotated chloroplast genome (MZ274135.1[72]) using Bowtie2[73]). The variant call considered all mapped reads without filtering and was performed using NGSEPcore[74]. A variant call file containing only biallelic loci was generated, and further analyses were carried out using R version 4.1.0[75]. The detected variants underwent further filtering, including a minimum sequencing depth of three reads, a maximum depth of 250, and individuals with more than 50% missing data, as well as SNPs with more than 25% missing data were removed (SNPfiltR package[76]). This resulted in a genetic dataset of 234 trees.

## Stable isotope analysis

Between two and 10 trees per site were selected for stable isotope analysis, depending on the isotope: we measured $\delta^{34}S$ in two to four trees per species per site (total of 46), $\delta^{2}H$ in four to 10 (total of 90) and $\delta^{18}O$ in six to 10 (total of 96) trees per site. $\delta^{13}C$ was measured in 12 sites, but was not found to add to the identification accuracy (results not shown). As measurements were missing at one site, this isotope was not included in the final model. Trees were selected for isotopic measurements such that multiple isotopes were measured for the same set of trees as much as possible. This resulted in a geolocated database of 96 Azobé trees in total, with between 1 and 3 isotopes measured per tree. This is the smallest dataset out of the three tracing models, because it was expected that local variation was smaller in the isotope ratios and the number of trees measured at short distances is in line with other studies that showed good tracing results[27,34]. The smaller dataset may have influenced tracing results; however, if the subset of samples did not characterize the site well.

Stable isotope measurements followed Boeschoten et al.[37]. In short, a subsample of heartwood from every tree was cut in radial direction, including at least 3–5 cm to include wood formed during multiple years. $\delta^{18}O$ and $\delta^{2}H$ were measured in cellulose, following Vlam et al.[19] for cellulose extractions. $\delta^{34}S$ was measured in whole wood. The stable isotopes ratios were determined by IRMS, expressed in per mill (‰) relative to an international reference standard (V-SMOW for $\delta^{18}O$ and $\delta^{2}H$ and CDT for $\delta^{34}S$).

## Multi-element analysis

15 to 20 trees per site were selected for multi-element analysis. The wood chemical composition was measured following Boeschoten et al.[18]. In short, a 1.0 g subsample was cut from 3–5 cm of heartwood and dissolved in 70% $HNO_3$ by heating in a microwave digestion system (Mars 6, CEM Cooperation, USA). The lowest detection limit per element was calculated as three times the intensity of that element in a blank standard. If trace elements were found in quantities below the detection limit in more than 100 samples (the equivalent of about half of the trees), they were excluded. This resulted in a multi-elemental composition of 41 trace elements, measured in 234 trees: Li, Na, Mg, Al, Si, P, K, Ca, Ti, Cr, Mn, Fe, Co, Ni, Cu, Zn, Ga, As, Rb, Sr, Y, Zr, Mo, Cd, Sn, Cs, Ba, La, Ce, Pr, Nd, Sm, Eu, Gd, Tb, Dy, Er, Yb, W, Pb, Bi. A summary of the number of samples included per method and their overlap is provided in Supplementary Table S2 and Fig. S1.

## Statistical analysis

All statistical analyses were performed in R version 4.2.3[77]. To test the variation between sites for each of the forensic methods, a db-RDA was performed based on 1-proportion of shared alleles for the pDNA (PopGenReport package[78]) and on Chord distances for stable isotope ratios and multi-element analysis (vegan package[79]).

## Site identification

Random Forest models were developed to identify site origins using the ranger package[80]. As Random Forest algorithms require complete datasets, missing values were imputed prior to analysis. These gaps primarily arose due to varying numbers of trees sampled per method (see Supplementary Table S2 for the sample size per method per site and Fig. S1 for the overlap between methods).

For the multi-element and genetic datasets, between 15 and 20 trees were measured per site, totalling 234 trees. Some SNPs were missing as a result of prior filtering steps, with these missing data points randomly distributed across both pDNA and individual trees. To address this, the most common allele at the population of origin was used to impute missing SNPs. While this approach may slightly overestimate model accuracy, since imputation could cause some trees to share identical SNP profiles, the impact on site identification was considered minimal.

For the stable isotope dataset, fewer individuals were measured per site (ranging from two to ten per isotope). To harmonize dataset sizes, isotopic values were imputed with site means for trees that had genetic and elemental data but lacked isotope measurements. The number of trees with imputed data differed for the test and the training datasets. To select a test dataset, a subset of test trees with complete elemental, genetic, $\delta^{2}H$ and $\delta^{18}O$ data was randomly selected and excluded from the training dataset to minimize imputation bias in model assessment (41 test trees, representing 50% of the 83 trees with at least two isotope measurements, distributed across all 13 sites). The test sets included some imputed $\delta^{34}S$ values, as this isotope was only measured in 46 trees. Consequently, the use of site mean $\delta^{34}S$ values in the test set may have led to a slight overestimation of isotopic model accuracy due to potential site-driven assignments. For the training datasets, imputed site means were used for $\delta^{2}H$, $\delta^{18}O$, as well for $\delta^{34}S$ to match the genetic and elemental datasets. This imputation may have reduced isotope-based identification accuracy if across-tree variation in isotopic values is high and therefore the site-mean is uncertain.

We selected 25 random sets of 41 test trees, after which we developed Random Forest models with the remaining training dataset (234 trees minus the 41 test trees). The test trees were assigned to their most likely origin based on seven types of Random Forest models based on different reference datasets: one model per forensic method (SNP, ISO, EL), three models for pairwise method combinations (SNP-ISO, SNP-EL, ISO-EL), and one model incorporating all three methods (SNP-ISO-EL). This resulted in 7 × 25 identification models. Model identification accuracy was assessed as the percentage of correctly assigned test trees. For each model, the sites to which test trees were assigned were saved and distances between the actual and assigned sites were calculated. Identification accuracy and mismatches were visualized based on the 25 × 41 test trees per Random Forest model category. The average and standard deviation of the percentage of samples assigned to the correct or incorrect site were calculated by summarizing the results of the 25 random forest models per category.

## Site verification

To assess the application potential of these tracing methods, we simulated a verification scenario that might occur in a forensic context. We randomly assigned a false origin claim (from our set of 13 sites) to a random subset of 41 test samples, to match the size of the test set of the identification models. After removing the focal test tree from the dataset, we developed the Random Forest assignment models to assign the tree its most likely origin.

We focused on two key outcomes: (A) the percentage of origin claims that could be correctly verified and (B) the percentage of false claims that could be correctly rejected. To simulate a scenario where the claimed origin is correct (A), we tested whether assignment models correctly assigned the 41 trees to their real origin. In the training datasets for these models, we included reference trees from the true origin as well as all 12 other sites in the training dataset for the assignment model. This reflects a situation where samples can be collected from the claimed origin, so reference samples from that origin will then be represented in the reference database (see diagram in Fig. 4). Contrastingly, to simulate a scenario where the claimed origin is incorrect (B), we tested whether assignment models correctly rejected a false origin claim for the same 41 trees. For these models, we excluded all training trees of the true origin of the sample from the reference database when building the assignment models. The samples were then assigned to their most likely origin. If their assigned origin was different from the reported origin, a claim was reported as rejected. This represents a situation where the true origin is not the claimed origin, and moreover, the true origin is likely unknown and thus not present in the database. In this case, verification is thus based solely on the available reference sites, not including data from the claimed origin.

In these scenarios, we assume that someone related to the case can visit the claimed origin to collect samples of the same species at that location, which is possible in our study region but may not always be possible, for example, in conflict areas. Therefore, we acknowledge that this scenario does not cover all verification tests that may be needed in forensic cases. Nonetheless, we consider it a valuable assessment of how the tracing methods would perform in such a scenario and whether combining methods can improve verification success.

## Data availability

The genetic dataset is available in the Sequence Read Archive (SRA)/NCBI repository accession number: PRJNA1150388; and in the European Variation Archive (EVA) at EMBL-EBI under accession number PRJEB78866. The polymorphisms (SNPs) are available in the European Variation Archive (EVA) at EMBL-EBI under accession number PRJEB78866. Additionally, the full dataset supporting this publication, including the stable isotope ratios and multi-element concentrations, is published at Zenodo (https://doi.org/10.5281/zenodo.16107702). Please note that no commercial use is permitted under the current permits and any further use of these materials is subject to authorization from the relevant authorities in the countries where the biological material was collected. All materials used in this study were collected under the following permits: Cameroon- Research Permit No. 00000116/MINRESI/B00/C00/C10/C12 (Yaoundé, 09 Sep 2019); Research Permit No. 000066/MINRESI/B00/C00/C10/C12 (Yaoundé, 07 Jun 2021); Scientific research permit No. 2144 PRBS/MIN-FOF/ SETAT/SG/DFAP/SDVEF/SC/NGY (Yaoundé, 23 Jul 2021); ABS Permit 00010/MINEPDED /CNA/NPABS/ABS-FP (Yaoundé, 03 Dec 2021); PIC Decision No. 00013/D/MINEPDED/CNA of 03 Dec 2021. Gabon- Research authorization No. AR017/21/ MESRTTENCFC/CEN-AREST/CG/CST/CSAR.

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

## Acknowledgements

This study was supported by the Dutch Research Council (NWO-TTW-OTP-16427). Additional fieldwork support was received from the Alberta Mennega Foundation, FSC International and from World Forest ID. Additional analysis support was received from the Ecology Fund of the KNAW. We thank all the collaborating timber companies and their field teams for facilitating the fieldwork and all involved colleagues at our partner institutes, University of Dschang, Marien Ngouabi University, IRAF/CENAREST, the National Herbarium of Gabon, and IRET/CENAREST for their invaluable contributions.

## Author contributions

L.E.B., B.R.V.M.S., M.V., A.d.G., M.J.M.S., U.S. and P.A.Z. conceived the idea. L.E.B., B.R.V.M.S., A.B. U.G.B.D., J.C.U.C., N.L.E.O., M.G, J.J.L., D.M.F.M., C.G.M., R.M.D.N., D.N.B., S.N.T., M.T.T., B.B.L.T., H.T.Z. and P.T.Z. contributed to data collection. L.E.B. and B.R.V.M.S. analyzed the data and wrote the original draft. L.E.B., B.R.V.M.S., M.V., A.d.G., M.J.M.S., U.S. and P.A.Z. reviewed, commented, and edited the final manuscript. All authors reviewed the final manuscript.

## Competing interests

The authors declare no competing interests.
