## [Peer review file · Communications Earth & Environment]

Combined genetic and chemical methods boost the precision of tracing illegal timber in Central Africa

Corresponding Author: Dr Laura Boeschoten

This manuscript has been previously reviewed at another Nature Portfolio journal. This document only contains reviewer comments and rebuttal letters for versions considered at Communications Earth & Environment.

Version 0:

Decision Letter:

Dear Dr Boeschoten,

Your manuscript titled "Combining genetic and chemical methods boosts accuracy to trace illegal tropical timber" has now been seen by 2 reviewers, whose comments are appended below. You will see that they find your work of some potential interest. However, they have raised quite substantial concerns that must be addressed. In light of these comments, we cannot accept the manuscript for publication but would be interested in considering a revised version that fully addresses these serious concerns.

We hope you will find the reviewers' comments useful as you decide how to proceed. Should additional work allow you to

- address these criticisms (that is, either to incorporate the suggestions or provide a compelling argument why the point made by the reviewer is not valid or relevant to the editorial threshold as outlined below)

AND

- meet our editorial thresholds as outlined below,

then we would be happy to look at a revised manuscript.

In the following, we list requirements for publication.

*****Provide novel and fully supported insight into the accuracy of tracing illegal tropical timber.**

*****Outline your method and data in detail, including information on the size of concessions, explain sampling design, testing, and robustness; clarify the comparison and resolution of your approach over the others and address inconsistencies.**

******Expand the discussion of your findings, including importance within the African context, and demonstrate that all your data and analysis fully support your claims, alternatively, you must tone down your claims.**

When resubmitting, please provide a point-by-point response to the reviewers' comments. Please submit your responses as a separate file, distinct from your cover letter where you can add responses to the Editors' comments that you do not want to be made available to the reviewers. Word files are preferred. We recommend that any figures, tables or graphs that are included in the response to reviewers are also included in the main article or Supplementary Information.

If the revision process takes significantly longer than three months, we will be happy to reconsider your paper at a later date, as long as nothing similar has been accepted for publication at Communications Earth & Environment or published

elsewhere in the meantime.

Please use the following link to submit your revised manuscript, point-by-point response to the reviewers' comments with a list of your changes to the manuscript text (which should be in a separate document to any cover letter), a tracked-changes version of the manuscript (as a PDF file) and any completed checklist:

Link Redacted

Please do not hesitate to contact us if you have any questions or would like to discuss the required revisions further. Thank you for the opportunity to review your work.

Best regards,

Dushan Kumarathunge
External Editor
Communications Earth & Environment

Martina Grecequet, PhD
Senior Editor
Communications Earth & Environment

EDITORIAL POLICIES AND FORMAT

If you decide to resubmit your paper, please ensure that your manuscript complies with our editorial policies and complete and upload the checklist below as a Related Manuscript file type with the revised article:

Editorial Policy Policy requirements
(Download the link to your computer as a PDF.)

- Behavioural and social science
- Ecological, evolutionary & environmental sciences
- Life sciences

<https://www.nature.com/documents/nr-reporting-summary.zip>

For your information, you can find some guidance regarding format requirements summarized on the following checklist: (<https://www.nature.com/documents/commsj-phys-style-formatting-checklist-article.pdf>) and formatting guide (<https://www.nature.com/documents/commsj-phys-style-formatting-guide-accept.pdf>).

REVIEWER COMMENTS:

Reviewer #1 (Remarks to the Author):

This work by Boeschoten et al. is the culmination of a multi-year project called Timtrace, which investigates several methods for timber tracing in Central Africa, including stable isotope ratios, trace elements, and genetics. This is the first paper to combine all three methods, and the field has been eagerly awaiting its publication. The work is timely, and the authors are experts in their respective fields. This study represents an important advancement in method combination. However, there are some points that need to be addressed, both on language and on method development/comparison.

MAJOR COMMENTS

(1) There are three issues that make the author's "resolution" claim, and their claim that this outperforms the state-of-the-art, slightly misleading:
- The authors use fixed classification among 13 sites, but their primary argument for why these results are groundbreaking is partly based on kilometer accuracy. They repeatedly claim to achieve accuracies of < 50 km and < 100 km because the misassigned trees are classified to nearby sites from the actual original site. However, this is solely due to the distances

between their sites. This also ignores that their sites or concessions are quite large, so the distance might actually be bigger than what they claim. If we pick two extremely large concessions next to each other and we test trees as far away from the common border, they could claim a "0 km" resolution. Kilometer resolution can only be used when applying a spatial method, such as isoscapes, as demonstrated in Watkinson et al.'s "The Development and Use of Isoscapes to Determine the Geographical Origin of Quercus spp. in the United States" or Truszkowski et al.'s "A probabilistic approach to estimating timber harvest location." Even more importantly, we need to bear in mind that the kilometer accuracy depends on the size of the study area. The larger the area, the larger the error, so these numbers aren't comparable between studies anyway (unless it's 2 studies from roughly the same area).

- In a real-life testing scenario, we do not have samples from every concession/region of interest. Differentiating between two concessions where we do have samples (and can include them in the training set) will likely be much easier than differentiating between concessions where we don't have samples. So the results they report represent an idealized scenario that will not happen in practice. For verification, one can imagine sending collectors to a concession, but not for determination/origin identification, as you would have to send them everywhere. It is important to note this in the publication, as it might be taken at face value by enforcement and practitioners.

- They incorrectly state that, for stable isotope ratio analysis, "This technique is frequently applied in timber tracing but mostly yields high accuracy at large spatial scales (>500 km) [21, 29, 31]." -> There are several studies that have achieved much better results than >500 km (see a list further down). Additionally, the kilometer argument may not be the most effective when discussing classification models in your own work (see comment above).

(2) It appears that the single method comparison is based on different numbers of trees for each method: 322 trees for genetics, 295 trees for stable isotope ratio analysis, and 398 trees for multi-element analysis. In general, the methodology and sample division is hard to follow and could lead to confusion for readers. Perhaps I interpreted this wrong, but then I would advise the authors to make this clearer, or to make a Venn diagram that shows overlap for the combination of methods (see further down). If this discrepancy is real, then it could significantly impact the classification results. Perhaps if the authors used a minimal balanced dataset to perform the comparison.

OTHER COMMENTS

ABSTRACT

Line 35-36: While testing methods aim for high accuracy, the ultimate decision on whether the accuracy is sufficient for legal purposes is determined by (1) law enforcement finding it adequate to prosecute and (2) courts, considering the specific evidence and the standards set by the legal system. An assignment accuracy of 70% can be sufficient if there is additional evidence supporting the case. Moreover, other data techniques such as boxplots, trend dynamics, and PCA are also accepted in developing a case and moving to prosecution. It might be helpful to rephrase this sentence or consider removing the term 'forensics,' as it is often misused in the context of timber tracing methods."

Line 40: "It might be better to avoid the term 'unprecedented,' as other authors have achieved over 90% assignment accuracies for closely located populations or clusters. Your combination of methods is unprecedented, not necessarily the result.

Line 43: "If the method works well across the board, why limit it to high-value timbers and high-risk regions? Is this due to the cost of measuring all the variables?"

INTRODUCTION

Line 57: "A new Regulation on Deforestation-Free Products (EUDR, into effect in 2026) mandates importers and traders to ensure that timber is both legally sourced and has contributed to deforestation."

Suggestion: It seems there might be a slight oversight here. Perhaps you meant to say "has not contributed to deforestation"? Additionally, the regulation comes into effect on December 30, 2025.

Line 59: "from non-deforested areas"

Suggestion: It might be clearer to specify "from non-deforested areas post the cut-off date." This distinction is important within the legal framework.

Line 65: "the geographic of timber"

Suggestion: Geographic origin of timber?

Line 67-69: "Effective origin verification of suspicious timber shipments requires forensic methods that can independently identify timber origin with high accuracy at a relevant spatial scale for law enforcement."

Suggestion: Origin verification is not identifying timber origin with high accuracy. Origin verification is establishing whether there is a significant doubt on the claim of harvest location. As your work deals with origin identification, perhaps switch out origin verification. In addition, while identifying the exact origin can strengthen the case, it is not always a requirement. The focus is on verifying the claim made, and fraud occurs when this claim is incorrect.

Line 71: "These levels of accuracy and precision would enable the identification of many reported timber laundering routes,

such as timber from protected areas exported as if it comes from a legal concession, timber transported across borders to be sold from a second country, and wood extracted as part of forest conversion to other land uses [3, 17]."

Suggestion: It's worth noting that the claim of accuracy within 100 km might not always differentiate timber from protected areas versus legal concessions, especially when these areas are adjacent to each other, which is often the case. In addition, the country distinction can be quite challenging as well as logging can happen in border forests.

Line 74-76: "Yet, so far, none of the three most widely used methods - genetic analyses, stable isotope ratios, and multi-element analysis - has consistently demonstrated these levels of accuracy (>90%) and precision (< 100 km) to trace the origin of timbers [5, 6, 12, 18–21]."

Suggestion: This statement could use some nuance. For instance, the work cited later by Mortier et al. demonstrates that origin verification at the concession level across large spatial scales is possible. While this does not identify the exact origin, it aligns with the goals of casework, which focus on verifying the plausibility of origin claims rather than pinpointing the source.

Figure 1: It's worth noting that delta34S is also heavily influenced by bedrock so there is overlap with the trace elements drivers

Line 91-92: "This technique is frequently applied in timber tracing but mostly yields high accuracy at large spatial scales (>500 km) [21, 29, 31]."

Suggestion: There are several studies that have achieved much better results than >500 km. It might seem that the cited papers were selected to support the author's specific argument. Additionally, the kilometer argument may not be the most effective when discussing classification models in your own work (see major comment above). Even more importantly, we need to bear in mind that the kilometer accuracy depends on the size of the study area. The larger the area, the larger the error, so these numbers aren't comparable between studies anyway (unless it's 2 studies from roughly the same area). So to use this as a comparison as to why your method performs better is not sincere. Here are some examples of papers that obtained better than 500 km results:

Kagawa and Leavitt (2010): "Stable carbon isotopes of tree rings as a tool to pinpoint the geographic origin of timber"

Gori et al. (2018): "Timber isoscapes. A case study in a mountain area in the Italian Alps"

Watkinson et al. (2022): "Stable Isotope Ratio Analysis for the Comparison of Timber From Two Forest Concessions in Gabon" – 240 km

Mortier et al. (2024): "A framework for tracing timber following the Ukraine invasion"

Truskowski et al. (2025): "A probabilistic approach to estimating timber harvest location"

Line 103-104: It might be beneficial to emphasize the unique strengths of your methodology for Africa, rather than focusing solely on the "90% - forensics – small spatial scales" argument. This could make your message stronger and more compelling.

Line 121-127: It is appreciated that the authors clarify the focus on identification. This point could be moved earlier in the text to clarify that the discussion in the introduction on method, paper, and assignment success/percentage relates to origin identification, not verification. This adjustment would address some of my earlier points and provide a more balanced comparison in the introduction.

Figure 2: This is a fantastic figure. However, you might consider increasing the resolution. It is a bit challenging to distinguish between black and blue. Perhaps using a lighter shade of blue could help.

MATERIALS AND METHODS

Line 297-...: Would it be possible to provide some information on the sizes of the concessions? It would help to better understand the spatial scale of the analysis.

Line 297-...: Also, just to clarify—are the sites located within the concessions, or are "site" and "concession" being used interchangeably here?

Line 302-303: If I understand correctly, it's possible that two trees from different concessions could be geographically closer to each other than to other trees within their own concession—for example, in East Cameroon? Were such cases associated with higher misclassification rates? Exploring this further could potentially support your argument regarding the importance of distance.

Line 329-332: "Haplotypes were defined in a non-restrictive manner with inclusion of sequences of varying lengths within the same haplotype, while ambiguities resulted in certain sequences being assigned to different haplotype" -> not clear what this means exactly

Line 334-336: The sample sizes in this section seem relatively small. It might be helpful to comment on how this could affect the robustness or interpretation of the results. As mentioned above, the sample size should be stated more clearly in general.

Line 337: I'm a bit confused about how you arrive at the total of 295 Azobé trees. If each measurement corresponds to a different tree, wouldn't the sum be $51 + 101 + 105 = 257$? Perhaps I'm missing something - could you clarify? + It is unclear

which trees have complete data across methods -> I would suggest a Venn diagram to show how many samples have a specific combination of measurements and which samples were chosen for testing.

Line 366-372: This paragraph needs more clarity.

- It states that test samples were randomly chosen from trees that had d2H and d18O measurements. What about d34S?
- 44 test samples were chosen, and that was half of all those that had d2H and d18O measurements? Or all those that had all the isotopic measurements? Or all those that had the full set of isotopic, elemental and genetic measurements? Is it true that there were 88 complete samples in total?

Lines 379-382: 25 x 44 trees would indicate that there were 1100 test trees in total, which is not true. I suspect that there were 25 train/test splits, each of which corresponding to a different random choice of 44 trees out of the 88 trees that did not require data imputation. Is that correct? This needs to be stated more clearly.

DISCUSSION

Line 193: "Origin fraud in timber trade remains widespread and to date no tracing method has been able to consistently and reliably distinguish between origins at distances relevant for forensic cases (i.e, over 90% accuracy at <100 km). -> The current phrasing may slightly overstate the limitations of existing tracing methods. It might be more balanced to acknowledge ongoing advancements while emphasizing that no method has yet achieved consistent high accuracy (<100 km) across diverse contexts. Framing the importance of your methodology within the African context could actually make for a more compelling and grounded argument.

Line 206: "To put this into perspective, the average concession size in the Congo Basin is approximately 133 km². This could roughly translate to an area of 30 km by 45 km, depending on its shape" -> A 30km by 45km rectangle is roughly 1350 km², so around 10x the area reported. 133 km² corresponds to roughly 11km x 12 km, but distances will differ depending on what shape the concessions are. This should be fixed. (see also next comment)

Line 208: These are impressive results. That said, it's worth noting that they rely on having prior knowledge of all possible site options (i.e., a fixed classification approach). This doesn't fully address the more open-ended query of where a sample originates within a broader area. With just two known concessions, for instance, one might achieve close to 100% accuracy. Including this nuance would help clarify the scope and implications of the results.

Line 223: This sentence is a bit difficult to follow.

Line 227: "When compared to chemical tracing studies, our assignment success when combining methods far exceeds that of any study at the scale of <100 km." -> This comparison is a bit unfair and might come across as strong. It could be helpful to qualify the statement to reflect differences in scale, context, and methodology between your study and previous chemical tracing studies.

Line 233–238: It is important to note that the Mortier et al. study was conducted over a broad spatial scale across Eastern Europe and did not use a classification-based approach. Including this context would help readers better interpret the comparison between the 180–230 km accuracy range and your findings at <100 km.

Line 256: "To improve isotopic origin identification, the addition of carbon and strontium isotopes could further improve tracing results [25,58]." -> You mention here that adding carbon and strontium isotopes could enhance tracing results. However, earlier in the methods you indicate that carbon did not significantly improve assignment accuracy. It might be worth clarifying why carbon is still being recommended, or under what circumstances it might add value.

Reviewer #2 (Remarks to the Author):

Very interesting aim of identifying the origin as opposed to frequently investigated origin verification.

Comments for revision:

1. Use of GPS coordinates to places would help to easily locate the sample sites in future.
2. L88- What is Figure 1.1?
3. And you say (L99-L103) it is heavily traded, how and why? May be add clear uses of this species to biodiversity or as a commercial commodity used to make wooden furniture etc.
4. the last bit of introduction (L124-L126) you are trying to make a point about the significance of identifying. However, the argument is not clear. Please rephrase.
5. For a reader outside, botanical background, it would be better if you can add details of this tree species including pictures of the tree and timber.
6. In figure 1.1- What is the correlation between shapes of the three maps? Aren't they showing the same region? If so why different zoom levels? And please add legends to describe colour usage. What is yellow and orange in the third imagen of the map?
7. L140-142- So if values vary within sites how did you assign a value with an error percentage? May be provide a site specific GRF values without giving a very large 0-100% range.
8. L143- so only CON2 and CON4?

9. You also need to provide how these sites were picked and naming given (non-abbreviated version)
10. L155-:156- Can you explain what you observed as spatial structure? Please add an image to show this point. A table with sites and assignment accuracies would fit better for clear identification.
11. L159- Why and how did you pick trace elements?
12. L161- what were these "few trace elements"?
13. L68- shorter as in height? You did not compare dimensions before. So why now? Please be consistent what type of results you compare.
14. L171- What did you use for multi-element analysis?
15. Figure 2 In the caption you say "Colours in the outer circle indicate to which site the trees of that location were assigned". So needs a legend to identify colours.
16. L216- Have you consider assessing these soil properties? The list of trace elements you selected does it include elements present in soil? Basically, these sites might be the main contributing factor for differences among sites.
17. L217-218- Please rephrase.
18. L241-L245- Suggest re-phrasing without undermining your main goal of using multiple methods.
19. L256-258- Is its possible to cover the whole area? Cannot take representative samples?
20. L284- Suggest re-phrasing without undermining your main goal of using multiple methods to increase accuracy of prediction.
21. This included transboundary sample collection. Has there been necessary steps to obtain permits?
22. Any data on DNA quality and quantity? Cambium isn't an easy sample to extract DNA.

Communications Earth & Environment is committed to improving transparency in authorship. As part of our efforts in this direction, we are now requesting that all authors identified as 'corresponding author' create and link their Open Researcher and Contributor Identifier (ORCID) with their account on the Manuscript Tracking System prior to acceptance. ORCID helps the scientific community achieve unambiguous attribution of all scholarly contributions. You can create and link your ORCID from the home page of the Manuscript Tracking System by clicking on 'Modify my Springer Nature account' and following the instructions in the link below. Please also inform all co-authors that they can add their ORCIDs to their accounts and that they must do so prior to acceptance.

Version 1:

Decision Letter:

Dear Dr Boeschoten,

Your manuscript titled "Combining genetic and chemical methods boosts accuracy to trace illegal tropical timber" has now been seen by our reviewers, whose comments appear below. In light of their advice we are delighted to say that we are happy, in principle, to publish a suitably revised version in Communications Earth & Environment.

We therefore invite you to revise your paper one last time to address the remaining concerns of our reviewers. At the same time we ask that you edit your manuscript to comply with our format requirements and to maximise the accessibility and therefore the impact of your work.

EDITORIAL REQUESTS:

****Please take care to match our formatting and policy requirements. We will check revised manuscript and return manuscripts that do not comply. Such requests will lead to delays. ****

SUBMISSION INFORMATION:

OPEN ACCESS:

Communications Earth & Environment is a fully open access journal. Articles are made freely accessible on publication. For further information about article processing charges, open access funding, and advice and support from Nature Portfolio, please visit <https://www.nature.com/commsenv/open-access>

Link Redacted

Best regards,

Dushan Kumarathunge, PhD
Editorial Board Member
Communications Earth & Environment

Martina Grecequet, PhD
Senior Editor,
Communications Earth & Environment
Consulting Editor
Communications Sustainability

REVIEWERS' COMMENTS:

Reviewer #1 (Remarks to the Author):

I heavily appreciate the authors edits, and their counter-arguments for when they did not agree with my comments.

Four minor comments -

(1) in the verification piece, in scenario B, it is unclear on what "rejection" is based.

-> As I understand you do this based on whether the tree is assigned to a site that is not the true location (as it is removed), but also not the false location (as that site is still in the model)?

-> You also write "11 out of 12 samples", but in the text and schematic you talk about 41 trees. Perhaps I misunderstood, but maybe check whether the numbers are correct in this section?

(2) Line 288 -> change high "verification" accuracy to high "determination" accuracy. In Mortier et al, the verification test has no distance dimension to it and has a different implementation than the determination test. The 180-230 km is only for the determination question. Otherwise this section is good, and much appreciated on the clarification of methods and study comparison (also in other parts of the text)

(3) Line 328 -> I think a space is missing between accuracy and [28]

(4) Line 391 -> I think a space is missing between package and [81]

Great work!

Reviewer #2 (Remarks to the Author):

Thank you for extensive edits. My minor comments are attached below

** Visit Nature Portfolio's author and referees' website at www.nature.com/authors for information about policies, services and author benefits**

Dear editors,

Thank you for considering our manuscript 'Combining genetic and chemical methods boosts accuracy to trace illegal tropical timber' for publication in Communications Earth and Environment. We were happy to receive the opportunity to revise the manuscript based on comments by the editors and reviewers. We feel that addressing these comments has substantially improved the content and clarity of the paper. Please find a detailed response to your requests and reviewers' comments below. Line numbers in this response refer to the numbers in the track-changes Word document.

Best wishes,

Laura Boeschoten, Barbara RV Meyer-Sand, and co-authors

From the Editors:

***Provide novel and fully supported insight into the accuracy of tracing illegal tropical timber.

In our study, we provide the following novel insights into the accuracy of tropical timber tracing:

- (1) We present the first timber tracing study combining genetic and chemical methods. Doing so, our study fills an important knowledge gap in the field (Lowe et al. 2016 Bioscience, Dormontt et al 2015 Biological Conservation and Low et al. 2022 IAWA).
- (2) Novel and unprecedented are the levels of accuracy and precision obtained when combining methods. These have not been reached before for any tropical timber across a large scale and number of sites. We now better contextualize our results, by an extended analysis of literature, including more studies for other regions, timbers and tracing methods (L 110-131; L 322-342).
- (3) Insights on the complementarity across tracing methods are also clearly novel. We show that regions where one method performed poorly, others did better. This forms the basis of the high levels of correct assignment at small spatial scales.

These novel insights are fully supported by:

- (1) Small uncertainties of identification model outcomes. We now report model uncertainties not just for site assignment but also for distances of misassigned samples (new supplementary Figure S5).
- (2) A new origin verification analysis. As suggested by one of the reviewers, origin verification will be more important than determination in the practice of timber forensics. We therefore added a new origin verification analysis, which confirms that combining methods increases accuracy (new Figure 4).

***Outline your method and data in detail, including information on the size of concessions, explain sampling design, testing, and robustness; clarify the comparison and resolution of your approach over the others and address inconsistencies.

We have outlined methods and data in more detail. We now:

- 1) include more information on the size of the research sites. We realized the term 'concession' has caused confusion. This was not the scale at which any of the tests were performed. Rather, we used the word concession to indicate the geographic location of the site and to indicate we collaborated with local concessionaires. We did not suggest our sampling is representative for the entire concession, also because we sampled in those parts ('blocks') of the concession where logging operations would take place. Thus, other than as a name for the study location, the term concession has no meaning in the study. We therefore replaced 'concession' by 'site' throughout the manuscript. We also added text in the methods to explain this, and the size of a site, in more detail (L 421-431).
- 2) explain our sampling design more clearly in additional text when explaining the statistical tests (L 508-542) and in a Venn diagram that visualizes samples sizes per method (supplementary Figure S1) as well as a table with the number of trees measured per site (supplementary Table S2).
- 3) explain the statistical tests more clearly in text (L 508-542)
- 4) demonstrate robustness of our results across all distance bins, showing small levels of uncertainty resulting from different models (supplementary Figure S5).

We have clarified and addressed inconsistencies in the comparison and resolution of our approach compared to other studies by:

- 1) adding additional references in the introduction (L 110-131) to better support our claim that for tropical timber the application of individual tracing methods did not consistently yield high accuracy.
- 2) adding notes about the complexity of comparing results obtained from different species, geographical ranges, tracing methods and statistical approaches. We have added nuance when comparing to studies that were based on different statistical methods so that readers have more background to interpret the varying ways of defining resolution and we have added more information on the spatial scales (i.e. resolution) of other studies (L 322-342).
- 3) clarifying our use of the terms resolution, precision and accuracy (L 92-95), also in our response to reviewer 1 below. We acknowledge that in our analysis, misassignments will always be to other sites in our dataset. Therefore, possible distances of misassignments are determined by our study design. We stress that our study included a large range of between-site distances (10-1000 km), and that by reporting the distances at which misassignments occur (Fig 3), we provide full clarity and transparency about the spatial scales at which tracing methods operate. Our results show that misassignments tend to occur between geographically proximate sites, indicating that tracing errors do not simply increase with study area size but are influenced by underlying spatial variation

and the method used. We believe that this shows variation in the resolution of the different methods.

****Expand the discussion of your findings, including importance within the African context, and demonstrate that all your data and analysis fully support your claims, alternatively, you must tone down your claims.

To address this suggestion and expand the discussion of our findings:

(1) We have highlighted the relevance of our study to the African context, based on the arguments that illegal timber trade is widespread and the environmental variability is relatively small, which limits opportunities for tracing. We have added this context to the abstract (L38-41) and introduction (L59-60), allowing results to be interpreted with this background in mind. However, based on our findings of complementarity across methods, we are confident that combining tracing methods has the potential to improve resolution in other regions as well. Therefore, we do suggest testing combining tracing methods also for other timbers and in other regions (L 405-415).

(2) We have expanded our results and discussion by including an additional analysis that simulates a verification scenario (L 250-277 and Methods L 541-561, Figure 4). While the outcomes of this analysis are consistent with our identification tests, we believe it adds practical value by demonstrating how our findings can be applied in real-world verification contexts.

Based on the original and the additional analyses, we are confident that our data and analyses fully support the claims made in the manuscript.

REVIEWER COMMENTS:

Reviewer #1 (Remarks to the Author):

This work by Boeschoten et al. is the culmination of a multi-year project called Timtrace, which investigates several methods for timber tracing in Central Africa, including stable isotope ratios, trace elements, and genetics. This is the first paper to combine all three methods, and the field has been eagerly awaiting its publication. The work is timely, and the authors are experts in their respective fields. This study represents an important advancement in method combination. However, there are some points that need to be addressed, both on language and on method development/comparison.

Thank you for your interest! We were happy with your suggestions to improve the quality and clarity of the paper and we think it has significantly improved through this revision.

MAJOR COMMENTS

(1) There are three issues that make the author's "resolution" claim, and their claim that this outperforms the state-of-the-art, slightly misleading:

- The authors use fixed classification among 13 sites, but their primary argument for why these results are groundbreaking is partly based on kilometer accuracy. They repeatedly claim to achieve accuracies of < 50 km and < 100 km because the misassigned trees are classified to nearby sites from the actual original site. However, this is solely due to the distances between their sites. This also ignores that their sites or concessions are quite large, so the distance might actually be bigger than what they claim. If we pick two extremely large concessions next to each other and we test trees as far away from the common border, they could claim a "0 km" resolution. Kilometer resolution can only be used when applying a spatial method, such as isoscapes, as demonstrated in Watkinson et al.'s "The Development and Use of Isoscapes to Determine the Geographical Origin of *Quercus* spp. in the United States" or Truszkowski et al.'s "A probabilistic approach to estimating timber harvest location." Even more importantly, we need to bear in mind that the kilometer accuracy depends on the size of the study area. The larger the area, the larger the error, so these numbers aren't comparable between studies anyway (unless it's 2 studies from roughly the same area).

We appreciate the reviewer's comments and apologize for any confusion regarding our use of the term "concession". In our study, we sampled 20 trees per concession; however, these samples were not evenly distributed throughout each concession, and did not cover the entire concession. Instead, sampling was restricted to a site with a maximum radius of 5 km within each concession, ensuring that no two sampled trees within a single concession were more than 5 km apart (as described in the Methods). We recognize that referring to these areas as "concessions" may have caused confusion. To address this, we have revised the manuscript and now use the term "site" instead of "concession" consistently. We also added clarifying text to the Methods section to explain this distinction (L 419-429).

When testing for differences between assigned and actual origins (i.e., the spatial resolution at which we can reliably distinguish sites), we used the centroid of the latitude and longitude coordinates of all sampled trees within a site to represent its location. Distances between sites were then calculated based on the distances between these centroids. We thus did not run into the issue of claiming a zero-distance between trees that were actually sampled in far-away corners of two neighboring concessions.

In our analysis, we used sites (located within a forest concession) as the unit of tracing. We acknowledge that our distance estimates are therefore inherently linked to our sampling strategy and are thus defined by the distribution of between-site distances represented in our dataset. Nonetheless, we deliberately selected our research sites to encompass a broad range of distances, from as close as 10 km to as far away as 1010 km. This is illustrated in the 'RANDOM' column of Figure 3, where one can appreciate that pairwise distances between sites are distributed across various distance bins. We believe that reporting the distances at which trees are mis-assigned is informative, as it highlights the spatial scales relevant to the different methods we evaluated. Evidently, this is the case for all tracing studies performing site identification tests.

To enhance clarity, we have added a supplementary figure based on Figure 3 in which the bars are not stacked, allowing us to include error bars for each distance bin. Including all error bars in the main figure would reduce readability, but the supplementary figure provides readers with a clear visualization of the variation in assigned distances across different runs of the assignment models. The distribution of pairwise distances within our dataset is also more clearly depicted in the 'RANDOM' facet of this supplementary figure.

We recognize that our approach to spatial resolution differs from studies that use spatial prediction maps, as referenced by the reviewer. The resolution in our study may change if new sites are added to the database, but we note that this is also true for spatial prediction methods. In that case, adding more samples to the reference database also changes the prediction surface and thus the spatial resolution or precision. We have chosen to retain the term 'spatial resolution' in our manuscript because we show that the tracing methods differ in terms of the spatial scales at which misassignments occur, which can be effectively referred to as 'resolution' for tracing.

Finally, we respectfully disagree with the suggestion that error increases with the size of a study area. In our view, this relationship depends on both the tracing method employed and the underlying spatial variation. For example, expanding the study area to include a mountain range with distinct isotopic signatures could actually improve tracing accuracy. Our results support this: particularly in the genetic analysis, misassignments tend to occur between geographically closer sites rather than more distant ones. Thus, including more distant sites does not necessarily reduce accuracy.

- In a real-life testing scenario, we do not have samples from every concession/region of interest. Differentiating between two concessions where we do have samples (and can include them in the training set) will likely be much easier than differentiating between concessions where we don't have samples. So the results they report represent an idealized scenario that

will not happen in practice. For verification, one can imagine sending collectors to a concession, but not for determination/origin identification, as you would have to send them everywhere. It is important to note this in the publication, as it might be taken at face value by enforcement and practitioners.

This is an important point and we thank the reviewer for bringing this up. We definitely agree that we will never have samples from every potential origin, which is true for all forensic tracing studies and initiatives that aim to assemble large-scale databases (such as World Forest ID). However, when testing new tracing methods and aiming to understand under which circumstances they perform well, one needs to start with a certain set of locations and test for differences between these sites to show whether there is any basis for tracing purposes. This has and still is the basis for assessing method performance in the field of timber tracing. We considered identification models to be the most appropriate for this, as they give information on correct assignments but also yield distances of misassignments.

We did acknowledge this distinction between verification and identification early on in the paper: in the introduction we describe the difference and highlight our choice for verification, as is also acknowledged by the reviewer in a later comment. To further highlight our main focus on identification, we have added the term in the abstract (L 43) and we have changed wording throughout the manuscript from 'assignment accuracy' and 'assignment testing' to 'identification'.

The reviewer's comment also sparked an idea to perform an additional verification analysis using our Random Forest model approach. To evaluate to what extent combining methods also improves the performance of verification tests, we performed an additional set of tests. To this end, we took a subset of 41 trees (same size as the identification models) and randomly assigned origin claims from our dataset to each of these. We then verified these claims by assigning the trees to a potential origin. This is explained in full detail in methods (L 542-562) and results are included in a new Figure 4 and in the text (L 250-277). This analysis confirms that combining methods improves verification success, mostly by improving the % of correctly confirmed claims, and shows how our reference dataset and statistical models can be used to verify origin claims in a real word-example, if samples are obtained from the claimed origin.

- They incorrectly state that, for stable isotope ratio analysis, "This technique is frequently applied in timber tracing but mostly yields high accuracy at large spatial scales (>500 km) [21, 29, 31]." -> There are several studies that have achieved much better results than >500 km (see a list further down). Additionally, the kilometer argument may not be the most effective when discussing classification models in your own work (see comment above).

Thank you for the suggested papers, now included in the ms. We have changed this part of the introduction (L80-82) and give some background why we state that the potential for tracing with isotopes alone may be low in our study region specifically (L100-109). We have also highlighted our focus in the abstract (L 40-42), to emphasize the combination of methods is interesting especially when individual methods do not perform well enough, rather than giving the impression that individual methods never result in a high accuracy (which we did not intend to do).

(2) It appears that the single method comparison is based on different numbers of trees for each method: 322 trees for genetics, 295 trees for stable isotope ratio analysis, and 398 trees for multi-element analysis. In general, the methodology and sample division is hard to follow and could lead to confusion for readers. Perhaps I interpreted this wrong, but then I would advise the authors to make this clearer, or to make a Venn diagram that shows overlap for the combination of methods (see further down). If this discrepancy is real, then it could significantly impact the classification results. Perhaps if the authors used a minimal balanced dataset to perform the comparison.

Thank you for suggesting to include a Venn diagram to clarify our sampling strategy. We've now added such a Venn diagram (Fig S1, L 484-486) and a table that shows the number of samples per method per site (Table S2).

In addition, this comment also prompted us to re-think and slightly adapt our analysis strategy to produce a more simple comparison. Our adaptation involved an update of the models with a slightly smaller sample size for genetic analyses. Previously, we used all trees for which genetic analysis were conducted (322 trees) as a starting point. We imputed any missing SNPs (which were randomly distributed) and imputed missing data for the other methods (both isotopes and elements) with site means. This approach had the disadvantage that the genetic-only statistical models were based on a larger number of measured trees than the other models. Instead, we now used only the trees for which we had overlapping genetic and elemental data (234 trees) and only imputed the missing SNPs and isotope measurements. This had only minimal effect on the model results (see updated figures in the ms) but improved the clarity and logic of the methods and reduced the number of trees with imputed data.

Finally, we also added some more discussion on the potential implications of data imputation (L 515-528). In short, we argue that imputation of stable isotope data may have caused us to slightly 'over-state' the accuracy of isotope tracing because the imputed site means may improve accuracy. Yet, given that the assignment accuracy based on isotopic data is low, we consider the possible effects to have been relatively small and not to have changed the overall results of the method comparison.

OTHER COMMENTS

ABSTRACT

Line 35-36: While testing methods aim for high accuracy, the ultimate decision on whether the accuracy is sufficient for legal purposes is determined by (1) law enforcement finding it adequate to prosecute and (2) courts, considering the specific evidence and the standards set by the legal system. An assignment accuracy of 70% can be sufficient if there is additional evidence supporting the case. Moreover, other data techniques such as boxplots, trend dynamics, and PCA are also accepted in developing a case and moving to prosecution. It might be helpful to rephrase this sentence or consider removing the term 'forensics,' as it is often misused in the context of timber tracing methods."

Thank you, we've rephrased this here (L40) as well as in the introduction (L92-95) to clarify the point that forensic studies may not always need high accuracy at high resolution. However, as the reviewer states in the comments, sometimes you do need this high resolution and high accuracy if a nature reserve close to a concession needs to be compared, but individual methods have not yet shown consistently high accuracy at that scale.

Line 40: "It might be better to avoid the term 'unprecedented,' as other authors have achieved over 90% assignment accuracies for closely located populations or clusters. Your combination of methods is unprecedented, not necessarily the result.

Noted. We've clarified this to be unprecedented for our study region at this scale.

Line 43: "If the method works well across the board, why limit it to high-value timbers and high-risk regions? Is this due to the cost of measuring all the variables?"

Yes. We go into this argument in the discussion but don't have space left in the abstract to explain it in more detail. We've reworded it to remove some of the emphasis and to better link it to the previous statement.

INTRODUCTION

Line 57: "A new Regulation on Deforestation-Free Products (EUDR, into effect in 2026) mandates importers and traders to ensure that timber is both legally sourced and has contributed to deforestation."

Suggestion: It seems there might be a slight oversight here. Perhaps you meant to say "has not contributed to deforestation"? Additionally, the regulation comes into effect on December 30, 2025.

Thank you! Indeed, we had forgotten quite a crucial word here. Changed in ms. L64-65

Line 59: "from non-deforested areas"

Suggestion: It might be clearer to specify "from non-deforested areas post the cut-off date." This distinction is important within the legal framework.

Thank you. Changed in ms. L67

Line 65: "the geographic of timber"

Suggestion: Geographic origin of timber?

Thank you. Changed in ms. L75

Line 67-69: "Effective origin verification of suspicious timber shipments requires forensic methods that can independently identify timber origin with high accuracy at a relevant spatial scale for law enforcement."

Suggestion: Origin verification is not identifying timber origin with high accuracy. Origin verification is establishing whether there is a significant doubt on the claim of harvest location. As your work deals with origin identification, perhaps switch out origin verification. In addition, while identifying the exact origin can strengthen the case, it is not always a requirement. The

focus is on verifying the claim made, and fraud occurs when this claim is incorrect.

Thank you. Changed in ms. L77-82

Line 71: "These levels of accuracy and precision would enable the identification of many reported timber laundering routes, such as timber from protected areas exported as if it comes from a legal concession, timber transported across borders to be sold from a second country, and wood extracted as part of forest conversion to other land uses [3, 17]."

Suggestion: It's worth noting that the claim of accuracy within 100 km might not always differentiate timber from protected areas versus legal concessions, especially when these areas are adjacent to each other, which is often the case. In addition, the country distinction can be quite challenging as well as logging can happen in border forests.

We are aware of the complex cases where claimed and real origin are close by. However, as stated above by the reviewer, not all forensic cases require this accuracy. We did want to define a cutoff above which we called our models 'good'. Many of the illegal trade examples in the cited studies show that being able to distinguish origin at a scale of 100 km would allow competent authorities to catch the fraud. We have removed the examples as they can indeed be misleading if focussing on edge cases. We now just cite these references. L91-95

Line 74-76: "Yet, so far, none of the three most widely used methods - genetic analyses, stable isotope ratios, and multi-element analysis - has consistently demonstrated these levels of accuracy (>90%) and precision (< 100 km) to trace the origin of timbers [5, 6, 12, 18–21]."

Suggestion: This statement could use some nuance. For instance, the work cited later by Mortier et al. demonstrates that origin verification at the concession level across large spatial scales is possible. While this does not identify the exact origin, it aligns with the goals of casework, which focus on verifying the plausibility of origin claims rather than pinpointing the source.

We would like to emphasize the nuance in our statement, particularly the use of the word 'consistently'. While we acknowledge that various methods have yielded good results in certain cases, we maintain our position that their performance is not consistently high across all published studies. Some examples: the genetic work by Ng et al. (2017, PLOS ONE) shows good distinction between east and west Malaysia, but not at smaller scales. We found similar results in our work using genetic tracing in Central Africa: we found some very distinct sites in Gabon, but there was less small-scale spatial variation in other regions (Rocha Venancio Meyer-Sand et al. 2025 BMC Plant Biology). Same for our work on multielement analysis (Boeschoten et al. 2023 ERL), where we show some sites are very distinct at a small scale, while in other areas the chemical signal is more regional than site-specific. And for isotopes we see a similar pattern: studies have shown promising results at small scales (such as the mentioned work by Mortier et al.), but at the same time Paredes Villanueva et al. (2022, Tree-Ring Research) showed that isotopes showed little potential for sub-country identification in Bolivia and Boeschoten et al. (2023, Forest Ecology and Management) came to the same conclusion in Central Africa.

This is why we aimed to test whether, in situations where individual methods may not achieve the required accuracy, combining multiple methods could enhance tracing performance (as

mentioned throughout this response). We have reworded parts of the introduction on isotopic studies to clarify this point (L 78-82). We have also highlighted this in the abstract, to emphasize the combination of methods is interesting especially when the performance of individual methods is insufficient.

Figure 1: It's worth noting that delta34S is also heavily influenced by bedrock so there is overlap with the trace elements drivers

Thank you for that suggestion, we have included it in the figure.

Line 91-92: "This technique is frequently applied in timber tracing but mostly yields high accuracy at large spatial scales (>500 km) [21, 29, 31]."

Suggestion: There are several studies that have achieved much better results than >500 km. It might seem that the cited papers were selected to support the author's specific argument. Additionally, the kilometer argument may not be the most effective when discussing classification models in your own work (see major comment above). Even more importantly, we need to bear in mind that the kilometer accuracy depends on the size of the study area. The larger the area, the larger the error, so these numbers aren't comparable between studies anyway (unless it's 2 studies from roughly the same area). So to use this as a comparison as to why your method performs better is not sincere. Here are some examples of papers that obtained better than 500 km results:

Kagawa and Leavitt (2010): "Stable carbon isotopes of tree rings as a tool to pinpoint the geographic origin of timber"

Gori et al. (2018): "Timber isoscapes. A case study in a mountain area in the Italian Alps"

Watkinson et al. (2022): "Stable Isotope Ratio Analysis for the Comparison of Timber From Two Forest Concessions in Gabon" – 240 km

Mortier et al. (2024): "A framework for tracing timber following the Ukraine invasion"

Truszkowski et al. (2025): "A probabilistic approach to estimating timber harvest location"

Thank you for the suggested papers. We have rephrased this part of the introduction (L 78-82). We have referred to the kilometre argument in the responses above, as well as to the point that accuracy decreases when a study area increases.

Line 103-104: It might be beneficial to emphasize the unique strengths of your methodology for Africa, rather than focusing solely on the "90% - forensics – small spatial scales" argument. This could make your message stronger and more compelling.

Thank you. Changed in ms. abstract (L38-41) and introduction (L59-60)

Line 121-127: It is appreciated that the authors clarify the focus on identification. This point could be moved earlier in the text to clarify that the discussion in the introduction on method, paper, and assignment success/percentage relates to origin identification, not verification. This adjustment would address some of my earlier points and provide a more balanced comparison in the introduction.

We've reworded our methods from 'assignment' to 'identification' throughout the manuscript and also included 'identification' in the abstract, to clarify our methodology. Additionally, we've added a verification analysis to the results to show how these models could also aid verification testing. However, we feel like pulling this paragraph forward in the introduction halts the flow, so we've kept it there.

Figure 2: This is a fantastic figure. However, you might consider increasing the resolution. It is a bit challenging to distinguish between black and blue. Perhaps using a lighter shade of blue could help.

Thank you. We've added a legend to improve readability. We wanted to use a color coding that is color blind friendly and also prints well in grey scale (from the scientific color palette package `scico`). We think the legend helps interpretation of the results, and we think the following figure 3 helps when assessing differences in pairwise assignments between methods.

MATERIALS AND METHODS

Line 297-...: Would it be possible to provide some information on the sizes of the concessions? It would help to better understand the spatial scale of the analysis.

See comment on scale above; we've changed the wording to emphasize that we are working with sites that were sampled in concessions, but we did not cover a full concession. (L425-427)

Line 297-...: Also, just to clarify—are the sites located within the concessions, or are “site” and “concession” being used interchangeably here?

See previous comment.

Line 302-303: If I understand correctly, it's possible that two trees from different concessions could be geographically closer to each other than to other trees within their own concession—for example, in East Cameroon? Were such cases associated with higher misclassification rates? Exploring this further could potentially support your argument regarding the importance of distance.

See previous comment. No, trees from two sites, sampled in two different concessions, were not closer to trees from a nearby site than to other trees in their own site.

Line 329-332: "Haplotypes were defined in a non-restrictive manner with inclusion of sequences of varying lengths within the same haplotype, while ambiguities resulted in certain sequences being assigned to different haplotype" -> not clear what this means exactly

We have removed this sentence and apologize, the haplotypes were not used in the final data analysis.

Line 334-336: The sample sizes in this section seem relatively small. It might be helpful to comment on how this could affect the robustness or interpretation of the results. As mentioned above, the sample size should be stated more clearly in general.

We've added more explanation in the ms on sample size and overlap, as well as discussion on how this may affect the interpretation of the results. L500-528

Line 337: I'm a bit confused about how you arrive at the total of 295 Azobé trees. If each measurement corresponds to a different tree, wouldn't the sum be $51 + 101 + 105 = 257$? Perhaps I'm missing something - could you clarify? + It is unclear which trees have complete data across methods -> I would suggest a Venn diagram to show how many samples have a specific combination of measurements and which samples were chosen for testing.

Thank you, Venn diagram is added, supplementary Fig S1.

Line 366-372: This paragraph needs more clarity.

- It states that test samples were randomly chosen from trees that had d2H and d18O measurements. What about d34S?

As has now been clarified in the methods, less trees were measured for d34S so we did include some trees with imputed site means for d34S in the test dataset.

- 44 test samples were chosen, and that was half of all those that had d2H and d18O measurements? Or all those that had all the isotopic measurements? Or all those that had the full set of isotopic, elemental and genetic measurements? Is it true that there were 88 complete samples in total?

Updated in ms. L529

Lines 379-382: 25 x 44 trees would indicate that there were 1100 test trees in total, which is not true. I suspect that there were 25 train/test splits, each of which corresponding to a different random choice of 44 trees out of the 88 trees that did not require data imputation. Is that correct? This needs to be stated more clearly.

Noted, updated in the ms. 25x44 referred to the number of random draws of the trees that did not require imputation for all variables but d34S. L521-525

DISCUSSION

Line 193: "Origin fraud in timber trade remains widespread and to date no tracing method has been able to consistently and reliably distinguish between origins at distances relevant for forensic cases (i.e, over 90% accuracy at <100 km). -> The current phrasing may slightly overstate the limitations of existing tracing methods. It might be more balanced to acknowledge ongoing advancements while emphasizing that no method has yet achieved consistent high accuracy (<100 km) across diverse contexts. Framing the importance of your methodology within the African context could actually make for a more compelling and grounded argument.

See comments on resolution; we felt like the nuance was there in the original ms and have added more context in the discussion to highlight this even more (L 320-334).

Line 206: "To put this into perspective, the average concession size in the Congo Basin is

approximately 133 km². This could roughly translate to an area of 30 km by 45 km, depending on its shape" -> A 30km by 45km rectangle is roughly 1350 km², so around 10x the area reported. 133 km² corresponds to roughly 11km x 12 km, but distances will differ depending on what shape the concessions are. This should be fixed. (see also next comment)

Thank you for spotting this! It was indeed a miscalculation. We've removed this, to avoid any confusion between concession and site.

Line 208: These are impressive results. That said, it's worth noting that they rely on having prior knowledge of all possible site options (i.e., a fixed classification approach). This doesn't fully address the more open-ended query of where a sample originates within a broader area. With just two known concessions, for instance, one might achieve close to 100% accuracy. Including this nuance would help clarify the scope and implications of the results.

We appreciate the comment and as mentioned previously, we have included the 'RANDOM' assignment column in Figure 3 for this purpose: so that a reader can appreciate that the models perform better than what a random assignment based on this selection of sites would have been. We also included suggestions for further work in the discussion (L 379-394), where we state that future development could include interpolation methods to move forward from this site by site assignment.

Line 223: This sentence is a bit difficult to follow.

Ok, we've rephrased.

Line 227: "When compared to chemical tracing studies, our assignment success when combining methods far exceeds that of any study at the scale of <100 km." -> This comparison is a bit unfair and might come across as strong. It could be helpful to qualify the statement to reflect differences in scale, context, and methodology between your study and previous chemical tracing studies.

We removed the 'far' and have provided more context in the following sentences. We also added sentences to the start of the paragraph to highlight that it remains hard to compare results between studies (which is also why we think this study is so valuable) (L 320-334).

Line 233–238: It is important to note that the Mortier et al. study was conducted over a broad spatial scale across Eastern Europe and did not use a classification-based approach. Including this context would help readers better interpret the comparison between the 180–230 km accuracy range and your findings at <100 km.

Noted and we've updated the ms. (L 320-334)

Line 256: "To improve isotopic origin identification, the addition of carbon and strontium isotopes could further improve tracing results [25,58]." -> You mention here that adding carbon and strontium isotopes could enhance tracing results. However, earlier in the methods you indicate

that carbon did not significantly improve assignment accuracy. It might be worth clarifying why carbon is still being recommended, or under what circumstances it might add value.

Fair point! We've added some text to highlight ¹³C may add value in other regions, because indeed in our region it did not add tracing accuracy but in other regions it has yielded better results (L 363).

Reviewer #2 (Remarks to the Author):

Very interesting aim of identifying the origin as opposed to frequently investigated origin verification.

Thank you for your review and we're glad the identifying angle was interesting.

Comments for revision:

1. Use of GPS coordinates to places would help to easily locate the sample sites in future.

We agree that that would help to locate sites – however, we deliberately kept this out of the paper as we did not want our paper to be directly linked to concessions or companies as this could be seen as if we were tracing their timber in an illegality framework.

2. L88- What is Figure 1.1?

We refer to part 1 in Figure 1, left part of the figure that refers to the SNP analysis. We have rearranged the numbers in the figure, we hope this clarifies.

3. And you say (L99-L103) it is heavily traded, how and why? May be add clear uses of this species to biodiversity or as a commercial commodity used to make wooden furniture etc.

Thank you for that suggestion. We have added some text in the ms. L141

4. the last bit of introduction (L124-L126) you are trying to make a point about the significance of identifying. However, the argument is not clear. Please rephrase.

In response to this comment as well as comments by Reviewer 1, we have updated this paragraph and added a verification analysis. L157-168

5. For a reader outside, botanical background, it would be better if you can add details of this tree species including pictures of the tree and timber.

Thank you for this suggestion. We feel like the species in question is an example we have chosen to apply these methods for, however, we feel like emphasizing it more by adding pictures would make the study feel like it is less generalizable to other species. Therefore we have decided not to include pictures. We would like to add here that there is a lot of material on the species to be found online, so this information is easily accessible if a reader would be interested.

6. In figure 1.1- What is the correlation between shapes of the three maps? Aren't they showing the same region? If so why different zoom levels? And please add legends to describe colour usage. What is yellow and orange in the third imagen of the map?

The three maps do show the same region in the same zoom level, therefore we are not sure what the reviewer refers to. They originate from different sources, therefore the resolution of the

data that is shown does differ between the three. The legends are included in text in the figure legend, we apologize yellow and pink were not included there. We have added them. L135-137

7. L140-142- So if values vary within sites how did you assign a value with an error percentage? May be provide a site specific GRF values without giving a very large 0-100% range.

The error percentage comes from different random forest models where we removed a different test subset (25 different subsets of test samples as described in the methods). This is explained in the legend of figure 3 and we've added text in the methods to clarify (L539). We feel like the range is important to mention, as it shows how the method performs perfect for some sites but completely wrong for others, which explains why the overall mean is just below 50%. The summary of the correct assignments is visually represented in Figures 2 and 3.

8. L143- so only CON2 and CON4?

Thank you for noting that. We've added 'such as', as there are more examples that can be found in Figure 2. L185

9. You also need to provide how these sites were picked and naming given (non-abbreviated version)

Noted, we've added text in the Methods. L425-427

10. L155-:156- Can you explain what you observed as spatial structure? Please add an image to show this point. A table with sites and assignment accuracies would fit better for clear identification.

Although we thank the reviewer for the comment, we feel like the visualization in Figure 2 is a better way of showing this because a table does not have the same quality of the map, where a site can be included at it's location on the map. We've added text in the results to highlight what to look for when we talk about spatial structure and we think this helps in interpreting the results. L198-199

11. L159- Why and how did you pick trace elements?

We aimed to measure as many elements as possible, using a protocol previously established in Boeschoten et al (2022). However, some elements only occur in such low concentrations in the wood that we could not measure them (as mentioned in the methods). We kept all elements that we could measure from the large screening that was performed, following Boeschoten et al. 2022. We have added the list in the methods (L 483).

12. L161- what were these "few trace elements"?

As we focused on the combination of these methods rather than the individual-method performance we did not go into much detail for each method alone, such as which elements were most important for tracing purposes. We refer to Boeschoten et al. 2023, ERL, where this is done in more detail. We have added this reference to the text to direct readers to more information if they are interested.

13. L68- shorter as in height? You did not compare dimensions before. So why now? Please be consistent what type of results you compare.

Shorter as in distance. We have edited the sentence to make this more clear. L212

14. L171- What did you use for multi-element analysis?

We are not entirely sure what explanation the reviewer is asking for here. We used all elements as measured in the multi-element analysis and we've added the list of individual elements in the methods. We think the confusion may come from the term 'single method', which may be interpreted as if we use one element per analysis. We have changed the term 'single method' to 'individual method' throughout the manuscript when we refer to one of the three tracing methods, we hope this clarifies.

15. Figure 2 In the caption you say "Colours in the outer circle indicate to which site the trees of that location were assigned ". So needs a legend to identify colours.

As is also noted in the legend, this is indicated by the color of the inner circle of each site. As this was probably not clear enough we've added an extra legend and moved site names from within the circle to above the circle, so that the color is more clearly visible.

16. L216- Have you consider assessing these soil properties? The list of trace elements you selected does it include elements present in soil? Basically, these sites might be the main contributing factor for differences among sites.

Thank you for this comment. We did not assess the effect of soil properties on elemental composition in this paper, because we wanted to highlight the method comparison (as mentioned above). We did do so in a previous paper (Boeschoten et al. 2022 STOTEN) and we have added that reference here.

17. L217-218- Please rephrase.

Edited the sentence in the ms. L311-313

18. L241-L245- Suggest re-phrasing without undermining your main goal of using multiple methods.

We've edited the sentence in the ms, but we do think it is important to state that we don't think we will always need to use the combination of methods, if individual methods are already enough. L346-348

19. L256-258- Is its possible to cover the whole area? Cannot take representative samples?

We don't want to suggest it is possible to cover the whole area, this would require a sampling effort we don't think is feasible in the nearby future. As suggested, we indeed think the methods could work for tracing origin by sampling representative samples across potential origins. However, as can be seen from the map in Figure 2, here we do need to acknowledge there are areas from which we know timber is harvested (such as central Gabon) where we do not yet have samples. Which is what we mention in this part of the discussion.

20. L284- Suggest re-phrasing without undermining your main goal of using multiple methods to increase accuracy of prediction.

See point 18. We think, also given the comments of reviewer 1, that it is important to highlight that we don't suggest that we should always combine all available methods for timber tracing. We hope the reviewer agrees that we have only lightly rephrased this sentence.

21. This included transboundary sample collection. Has there been necessary steps to obtain permits?

Yes, we refer the reviewer to the acknowledgements which include the obtained permit numbers.

22. Any data on DNA quality and quantity? Cambium isn't an easy sample to extract DNA.

The reviewer is correct in noting that cambium is a challenging tissue for DNA extraction, particularly for chloroplast DNA (cpDNA), which is generally more abundant and easier to recover from fresh leaf samples. Although cpDNA copy numbers are typically lower in cambium than in leaves, we emphasize that using cpDNA remains advantageous in this context due to its higher copy number per cell compared to nuclear DNA and its shorter fragment size, which facilitates recovery from partially degraded material. Importantly, rigorous filtering steps were applied to ensure high-quality sequencing data (see Methods and RV Meyer-Sand et al., 2025).

In detail, prior to library preparation, DNA quantity was assessed using a Qubit fluorometer, and DNA concentrations used for library dilution ranged from 12.4 to 226.0 ng/μl. Agarose gel electrophoresis showed smear patterns in most samples, consistent with a range of fragment sizes for most of our samples (e.g. Figure R1 for fragment size distribution). Very fresh collected samples (< 6 months), as the ones in the first row in Figure R1, yield long fragments (~10000bp). Additionally, below we provide a graphical overview of DNA quality across tissue types and protocols (Figure R2 and R3), as discussed in the general discussion (chapter 6) of Barbara Rocha Venancio Meyer-Sand's thesis (see reference list at the end). Note: the figures R2 and R3 include additional data of other manuscripts (and chapters of the thesis).

Figure R1: Agarose gel with fragment size distribution using "Smart ladder" (50– 10000 bp).

Figure 2: Effect of species (Azobe and Tali, x-axis), tissue (facets) and DNA isolation protocol (colour) on DNA concentration. Boxplots show median, interquartile range and extreme values. DNA concentration data are derived from multiple chapters, including Chapter 2 (Tali sapwood, leaf and cambium, using CTAB protocol), Chapters 3 and 5 (Azobé leaf and cambium, CTAB protocol) and Chapter 4 (cambium and heartwood, using PlantKit protocol). No Azobé sapwood samples were included (adapted from RV Meyer-Sand 2024).

Figure 3 – Box plots comparing a) species (*Lophira* spp vs *Erythrophleum* spp) ($p < 0.001$); b) DNA isolation protocol (CTBA vs Plant Kit) ($p < 0.05$); plant tissues c) cambium vs heartwood ($p < 0.001$) and d) cambium vs leaf ($p < 0.001$) (adapted from RV Meyer-Sand 2024).

References in this response

Laura E **Boeschoten**, Mart Vlam, Ute Sass-Klaassen, Barbara Rocha Venâncio Meyer-Sand, Ulfa Adzkie, Gaël D U Bouka, Jannici C U Ciliane-Madikou, Nestor L Engone Obiang, Mesly Guieshon-Engongoro, Joël J Loumeto, Dieu-merci M F Mbika, Cynel G Moundounga, Rita M D Ndangani, Dyana Ndiade Bourobou, Mohamad M Rahman, Iskandar Z Siregar, Steve N Tassiamba, Martin T Tchamba, Bijoux B L Toumba-Paka, Herman T Zanguim, Pascaline T Zemtsa, and Pieter A Zuidema. “A new method for the timber tracing toolbox: applying multi-element analysis to determine wood origin”. In: *Environmental Research Letters* 18.5 (2023)

Laura E. **Boeschoten**, Mart Vlam, Ute Sass-Klaassen, Barbara Rocha Venâncio Meyer-Sand, Arnoud Boom, Gaël U.D. Bouka, Jannici C.U. Ciliane-Madikou, Nestor Laurier Engone Obiang, Mesly GuieshonEngongoro, Joël J. Loumeto, Dieu-merci M.F. Mbika, Cynel G. Moundounga, Rita M.D. Ndangani, Dyana Ndiade Bourobou, Peter van der Sleen, Steve N. Tassiamba, Martin T. Tchamba, Bijoux B.L. Toumba-Paka, Herman T. Zanguim, Pascaline T. Zemtsa, and Pieter A. Zuidema. “Stable isotope ratios in wood show little potential for sub-country origin verification in Central Africa”. In: *Forest Ecology and Management* 544 (2023)

Chin Hong **Ng**, Soon Leong Lee, Lee Hong Tnah, Kevin Kit Siong Ng, Chai Ting Lee, Bibian Diway, and Eyen Khoo. “Geographic origin and individual assignment of *Shorea platyclados* (Dipterocarpaceae) for forensic identification”. In: *PLOS ONE* 12.4 (2017)

Kathelyn **Paredes-Villanueva**, G. Arjen de Groot, Ivo Laros, Jan Bovenschen, Frans Bongers, and Pieter A. Zuidema. “Genetic differences among *Cedrela odorata* sites in Bolivia provide limited potential for fine-scale timber tracing”. In: *Tree Genetics and Genomes* 15.3 (2019).

Barbara **Rocha Venancio Meyer-Sand**. “Timber tales: tracing the origin and species identity of African timbers using plastid genomes.” PhD thesis. Wageningen University and Research, 2024.

Barbara **Rocha Venancio Meyer-Sand**, Laura E. Boeschoten, Gaël U.D. Bouka, Jannici C.U. CilianeMadikou, G. Arjen de Groot, Nathalie de Vries, Nestor L. Engone Obiang, Danny Esselink, Mesly Guieshon-Engongoro, Olivier J. Hardy, Simon Jansen, Joël J. Loumeto, Dieu-merci M.F. Mbika, Cynel G. Moundounga, Dyana Ndiade-Bourobou, Rita M.D. Ndangani, Marinus J. M. Smulders, Steve N. Tassiamba, Martin T. Tchamba, Bijoux B.L. Toumba-Paka, Herman T. Zanguim, Pascaline T. Zemtsa, and Pieter A. Zuidema. “Unlocking the geography of Azobé timber (*Lophira alata*): revealing spatial genetic structure beyond species boundaries”. In: *BMC Plant Biology* 25.1 (2025)

Dear editors,

Thank you for your positive response to our manuscript. We reply to the last responses from the reviewers here and supplied two version of the manuscript, one with and one without track changes. The version with track changes is to show how we've addressed the comments. The version without track changes follows the formatting and policy requirements.

Best wishes,

Laura Boeschoten, Barbara RV Meyer-Sand, and co-authors

From the Editor:

Ok! We have done so.

Reviewer #1 (Remarks to the Author):

I heavily appreciate the authors edits, and their counter-arguments for when they did not agree with my comments.

Four minor comments -

(1) in the verification piece, in scenario B, it is unclear on what "rejection" is based.

-> As I understand you do this based on whether the tree is assigned to a site that is not the true location (as it is removed), but also not the false location (as that site is still in the model)?

We have added text in the methods to clarify this, it was indeed applied in this way.

-> You also write "11 out of 12 samples", but in the text and schematic you talk about 41 trees.

Perhaps I misunderstood, but maybe check whether the numbers are correct in this section?

Thank you for this comment, we have edited this part of the results to make the text more clear.

With 11 out of 12, we referred to the potential origin sites in the assignment model. Not the total number of randomly assigned trees. The updated version better reflects this.

(2) Line 288 -> change high "verification" accuracy to high "determination" accuracy. In Mortier et al, the verification test has no distance dimension to it and has a different implementation than the determination test. The 180-230 km is only for the determination question. Otherwise this section is good, and much appreciated on the clarification of methods and study comparison (also in other parts of the text)

Thank you for this clarification, updated in the ms.

(3) Line 328 -> I think a space is missing between accuracy and [28]

Updated in the ms.

(4) Line 391 -> I think a space is missing between package and [81]
Updated in the ms.

Great work!

Reviewer #2 (Remarks to the Author):

Thank you for extensive edits. My minor comments are attached below

Dear Authors,

First of all, I really appreciate massive alterations you've done. And I really liked this version of the manuscript.

1. Use of GPS coordinates to places would help to easily locate the sample sites in future. We agree that that would help to locate sites – however, we deliberately kept this out of the paper as we did not want our paper to be directly linked to concessions or companies as this could be seen as if we were tracing their timber in an illegality framework.

True. This where you need consent and ethical clearance. Were these obtained? Because the base of your title itself is “traceability”.

Yes, those were obtained and all companies supported us in the collection of the samples.

2. L88- What is Figure 1.1?

We refer to part 1 in Figure 1, left part of the figure that refers to the SNP analysis. We have rearranged the numbers in the figure, we hope this clarifies.

You have more than 3 sub-images in Figure 1. You need to label all those and discuss them in the text. Those symbols are not needed in this context as the story is in the maps, having an icon only complicates. Also, I suggest using ABCD for sub images and be consistent with it to avoid confusion- in figure 2 you do this nicely. Further that driven by table also mislead the images as text don't match images.

We have changed the layout of the figure to three rows and updated the label to improve clarity. We do so because the illustrations at the top of the figure simply visualize the methods but they don't need to be referred to separately.

5. For a reader outside, botanical background, it would be better if you can add details of this tree species including pictures of the tree and timber.

Thank you for this suggestion. We feel like the species in question is an example we have chosen to apply these methods for, however, we feel like emphasizing it more by adding pictures would make the study feel like it is less generalizable to other species. Therefore we have decided not to include pictures. We would like to add here that there is a lot of material on

the species to be found online, so this information is easily accessible if a reader would be interested

Not really. Purpose of this comment was if there are clear morphological differences we as scientists need to clarify why we need genetic and chemical analyses. Especially that is how you define the impact of needing new tools to reduce fraudulent practices when species cannot be identified traditionally.

Thank you for clarifying! As this species is easy to identify in the field (very distinct bark and leaves) and we do not focus on species identification, we chose not to add pictures. We acknowledge that this would be helpful though in papers that are focused on species identification.

6. In figure 1.1- What is the correlation between shapes of the three maps? Aren't they showing the same region? If so why different zoom levels? And please add legends to describe colour usage. What is yellow and orange in the third imagen of the map? The three maps do show the same region in the same zoom level, therefore we are not sure what the reviewer refers to. They originate from different sources, therefore the resolution of the data that is shown does differ between the three. The legends are included in text in the figure legend; we apologize yellow and pink were not included there. We have added them. L135-137

Actually, what you describe in the text isn't exactly shown in the figures. Eg- only rivers are shown in 1.1 as barriers but you talk about seed dispersal patterns and tree proximity in the text. Please try rephrasing this section to match the figure.

In the text we refer to the general patterns that can divide genetic populations but you are right that in this context, the only natural barrier present are the rivers. We want to keep the introduction broad to link it to published studies. Yet, in the discussion, we do discuss the lack of spatial barriers in more detail.

And I suggest using the standard practice of color gradients (i.e light blue → low clay % and dark blue → high clay %). Thanks adding pink and yellow.

We appreciate the suggestion but the distinct colors in the figure show the variation in soil clay more clearly than a gradient can.

15. Figure 2 In the caption you say "Colours in the outer circle indicate to which site the trees of that location were assigned ". So needs a legend to identify colours.

As is also noted in the legend, this is indicated by the color of the inner circle of each site. As this was probably not clear enough we've added an extra legend and moved site names from within the circle to above the circle, so that the color is more clearly visible.

Appreciate the new change. And yes it makes things clearer.

I am sorry I am still not 100% satisfied with the wordings in the legend. Also I suggest using more contrasting colors so under this resolution things are clearer.

As it was not described what exactly what wrong with the legend text, we were not 100% sure what to change. We've added 'for each site' to make the reference to 'inner circle' more clear.

18. L241-L245- Suggest re-phrasing without undermining your main goal of using multiple methods.

We've edited the sentence in the ms, but we do think it is important to state that we don't think we will always need to use the combination of methods, if individual methods are already enough. L346-348

If you think so you are basically telling your method is not needed. So I suggest not stating the individual methods are enough over and over. Telling that once is enough. And clarifying you are doing this ONLY to be used in instances where differentiation via single methods are insufficient is the way to go.

Even though we appreciate the comment and the support for our combination of methods, we do think it is important to discuss when the combination is warranted as this is the very first time these methods were combined. Therefore, we are convinced a paper like ours needs to include a discussion about where and under what conditions combining methods would be desirable to improve tracing success. We cannot and do not want to claim that methods always need to be combined in timber tracing.

And, in response to the comment "you are basically telling your method is not needed": this is certainly not the case. In our view, this last part of the Discussion clearly mentions under what conditions (for what regions, species, situations), combining methods is likely warranted and useful. That's by no means a claim that the method would not be needed; rather it's a nuanced piece of discussion on when it is needed.

21. This included transboundary sample collection. Has there been necessary steps to obtain permits?

Yes, we refer the reviewer to the acknowledgements which include the obtained permit numbers.

Please state that is a permit not a grant number.

Perhaps the confusion has been that permit numbers are not in the section "Acknowledgements", but in "Data and materials availability". The sentence introducing these numbers reads: 'All materials used in this study were collected under the following permits: ... ', so this should be rather clear to readers.

22. Any data on DNA quality and quantity? Cambium isn't an easy sample to extract

DNA Thanks for sending extensive information on DNA quality. SO basically you used cpDNA from leaves, cambium, sapwood, heartwood? And did you take samples last wells of the rows 2nd and 3rd in Figure R1? AD0122-1-AS0040-1 ; OH3036-1- JLD1071-1

Yes, we have collected cpDNA from all the materials listed. However, for this manuscript we only included the cambium and leaf samples (as mentioned in the Methods). We refer to the thesis of

Dr. B RV Meyer-Sand for more details on how the rest of the samples were used (for blind tests, among other things). The specific samples listed were actually herbarium samples not used in this study, but in other chapters of Dr. B RV Meyer-Sand thesis and in R.V. Meyer-Sand et al. (2025).

Barbara **Rocha Venancio Meyer-Sand**. “Timber tales: tracing the origin and species identity of African timbers using plastid genomes.” PhD thesis. Wageningen University and Research, 2024.

Barbara **Rocha Venancio Meyer-Sand**, Laura E. Boeschoten, Gaël U.D. Bouka, Jannici C.U. CilianeMadikou, G. Arjen de Groot, Nathalie de Vries, Nestor L. Engone Obiang, Danny Esselink, Mesly Guieshon-Engongoro, Olivier J. Hardy, Simon Jansen, Joël J. Loumeto, Dieu-merci M.F. Mbika, Cynel G. Moundounga, Dyana Ndiade-Bourobou, Rita M.D. Ndangani, Marinus J. M. Smulders, Steve N. Tassiamba, Martin T. Tchamba, Bijoux B.L. Toumba-Paka, Herman T. Zanguim, Pascaline T. Zemtsa, and Pieter A. Zuidema. “Unlocking the geography of Azobé timber (*Lophira alata*): revealing spatial genetic structure beyond species boundaries”. en. In: BMC Plant Biology 25.1 (2025), p. 315.

Figure R1: Agarose gel with fragment size distribution using “Smart ladder” (50– 10000 bp).

Very interesting aim of identifying the origin as opposed to frequently investigated origin verification.

Comments for revision:

1. Use of GPS coordinates to places would help to easily locate the sample sites in future.
2. L88- What is Figure 1.1?
3. And you say (L99-L103) it is heavily traded, how and why? May be add clear uses of this species to biodiversity or as a commercial commodity used to make wooden furniture etc.
4. the last bit of introduction (L124-L126) you are trying to make a point about the significance of identifying. However, the argument is not clear. Please rephrase.
5. For a reader outside, botanical background, it would be better if you can add details of this tree species including pictures of the tree and timber.
6. In figure 1.1- What is the correlation between shapes of the three maps? Aren't they showing the same region? If so why different zoom levels? And please add legends to describe colour usage. What is yellow and orange in the third imagen of the map?
7. L140-142- So if values vary within sites how did you assign a value with an error percentage? May be provide a site specific GRF values without giving a very large 0-100% range.
8. L143- so only CON2 and CON4?
9. You also need to provide how these sites were picked and naming given (non-abbreviated version)
10. L155-156- Can you explain what you observed as spatial structure? Please add an image to show this point. A table with sites and assignment accuracies would fit better for clear identification.
11. L159- Why and how did you pick trace elements?
12. L161- what were these "few trace elements"?
13. L68- shorter as in height? You did not compare dimensions before. So why now? Please be consistent what type of results you compare.
14. L171- What did you use for multi-element analysis?
15. Figure 2 In the caption you say "Colours in the outer circle indicate to which site the trees of that location were assigned ". So needs a legend to identify colours.
16. L216- Have you consider assessing these soil properties? The list of trace elements you selected does it include elements present in soil? Basically, these sites might be the main contributing factor for differences among sites.
17. L217-218- Please rephrase.
18. L241-L245- Suggest re-phrasing without undermining your main goal of using multiple methods.
19. L256-258- Is its possible to cover the whole area? Cannot take representative samples?
20. L284- Suggest re-phrasing without undermining your main goal of using multiple methods to increase accuracy of prediction.
21. This included transboundary sample collection. Has there been necessary steps to obtain permits?
22. Any data on DNA quality and quantity? Cambium isn't an easy sample to extract DNA.

Dear Authors,

First of all, I really appreciate massive alterations you've done. And I really liked this version of the manuscript.

1. Use of GPS coordinates to places would help to easily locate the sample sites in future.

We agree that that would help to locate sites – however, we deliberately kept this out of the paper as we did not want our paper to be directly linked to concessions or companies as this could be seen as if we were tracing their timber in an illegality framework.

True. This where you need consent and ethical clearance. Were these obtained? Because the base of your title itself is “traceability”.

2. L88- What is Figure 1.1?

We refer to part 1 in Figure 1, left part of the figure that refers to the SNP analysis. We have rearranged the numbers in the figure, we hope this clarifies.

You have more than 3 sub-images in Figure 1. You need to label all those and discuss them in the text. Those symbols are not needed in this context as the story is in the maps, having an icon only complicates. Also, I suggest using ABCD for sub images and be consistent with it to avoid confusion- in figure 2 you do this nicely. Further that driven by table also mislead the images as text don't match images.

5. For a reader outside, botanical background, it would be better if you can add details of this tree species including pictures of the tree and timber.

Thank you for this suggestion. We feel like the species in question is an example we have chosen to apply these methods for, however, we feel like emphasizing it more by adding pictures would make the study feel like it is less generalizable to other species. Therefore we have decided not to include pictures. We would like to add here that there is a lot of material on the species to be found online, so this information is easily accessible if a reader would be interested

Not really. Purpose of this comment was if there are clear morphological differences we as scientists need to clarify why we need genetic and chemical analyses. Especially that is how you define the impact of needing new tools to reduce fraudulent practices when species cannot be identified traditionally.

6. In figure 1.1- What is the correlation between shapes of the three maps? Aren't they showing the same region? If so why different zoom levels? And please add legends to describe colour usage. What is yellow and orange in the third imagen of the map?

The three maps do show the same region in the same zoom level, therefore we are not sure what the reviewer refers to. They originate from different sources, therefore the resolution of the data that is shown does differ between the three. The legends are

included in text in the figure legend; we apologize yellow and pink were not included there. We have added them. L135-137

Actually, what you describe in the text isn't exactly shown in the figures. Eg- only rivers are shown in 1.1 as barriers but you talk about seed dispersal patterns and tree proximity in the text. Please try rephrasing this section to match the figure.

And I suggest using the standard practice of color gradients (i.e light blue → low clay % and dark blue → high clay %). Thanks adding pink and yellow.

15. Figure 2 In the caption you say “Colours in the outer circle indicate to which site the trees of that location were assigned “. So needs a legend to identify colours.

As is also noted in the legend, this is indicated by the color of the inner circle of each site. As this was probably not clear enough we've added an extra legend and moved site names from within the circle to above the circle, so that the color is more clearly visible.

Appreciate the new change. And yes it makes things clearer.

I am sorry I am still not 100% satisfied with the wordings in the legend. Also I suggest using more contrasting colors so under this resolution things are clearer.

18. L241-L245- Suggest re-phrasing without undermining your main goal of using multiple methods.

We've edited the sentence in the ms, but we do think it is important to state that we don't think we will always need to use the combination of methods, if individual methods are already enough. L346-348

If you think so you are basically telling your method is not needed. So I suggest not stating the individual methods are enough over and over. Telling that once is enough. And clarifying you are doing this ONLY to be used in instances where differentiation via single methods are insufficient is the way to go.

21. This included transboundary sample collection. Has there been necessary steps to obtain permits?

Yes, we refer the reviewer to the acknowledgements which include the obtained permit numbers.

Please state that is a permit not a grant number.

22. Any data on DNA quality and quantity? Cambium isn't an easy sample to extract DNA

Thanks for sending extensive information on DNA quality. SO basically you used cpDNA from leaves, cambium, sapwood, heartwood? And did you take samples last wells of the rows 2nd and 3rd in Figure R1? AD0122-1-AS0040-1 ; OH3036-1- JLD1071-1